**Combining Electromagnetic Induction and Satellite-based NDVI Data for Improved**
**Determination of Management Zones for Sustainable Crop Production**
**Authors**
Salar Saeed Dogar[1]*, Cosimo Brogi[1], Dave O'Leary[2,3], Ixchel M. Hernández-Ochoa[4], Marco
Donat[5,6], Harry Vereecken[1], and Johan Alexander Huisman[1]
[1] Agrosphere Institute (IBG-3), Forschungszentrum Jülich GmbH, 52425 Jülich, Germany
[2] Hy-Res Research Group, School of Natural Sciences, Earth and Life, College of Science and Engineering, University
of Galway, Galway, Ireland
[3] Teagasc, Animal and Grassland Research and Innovation Centre, Moorepark, Fermoy, Ireland
[4] Institute of Crop Science & Resource Conservation (INRES), Crop Science Group, University of Bonn, 53115 Bonn,
Germany
[5] Leibniz Centre for Agricultural Landscape Research, 15374 Müncheberg, Germany
[6] Faculty of Landscape Management and Nature Conservation, University for Sustainable Development (HNEE),
16225, Eberswalde, Germany
*Corresponding Author (s.dogar@fz-juelich.de)
**Abstract**
Accurate delineation of management zones is essential for optimizing resource use and improving
yield in precision agriculture. Electromagnetic induction (EMI) provides a rapid, non-invasive
method to map soil variability, while the Normalized Difference Vegetation Index (NDVI)
obtained with remote sensing captures above-ground crop dynamics. Integrating these datasets
may enhance management zone delineation but presents challenges in data harmonization and
analysis. This study presents a workflow combining unsupervised classification (clustering) and
statistical validation to delineate management zones using EMI and NDVI data in a single 70 ha
field of the patchCROP experiment in Tempelberg, Germany. Three datasets were investigated: (1)
EMI maps, (2) NDVI maps, and (3) a combined EMI-NDVI dataset. Historical yield data and soil
samples were used to refine the clusters through statistical analysis. The results demonstrate that
four EMI-based zones effectively captured subsurface soil heterogeneity, while three NDVI-based
zones better represented yield variability. A combination of EMI and NDVI data resulted in three
zones that provided a balanced representation of both subsurface and above-ground variability.
The final EMI-NDVI derived map demonstrates the potential of integrating multi-source datasets
for field management. It provides actionable insights for precision agriculture, including optimized
fertilization, irrigation, and targeted interventions, while also serving as a valuable resource for
environmental modelling and soil surveying.

## 1 Introduction

Reliable and accurate agricultural management zones that capture within-field variability affecting crop development can play a pivotal role in sustainable agriculture. Management zones can be used in the context of precision agriculture to optimize farming practices, increase yields, and reduce the use of available resources (Gebbers and Adamchuk, 2010; Janrao et al., 2019). This is not only valuable for profit maximization (Adhikari et al., 2022), but is also vital to meet future climate change and food security challenges (Antle et al., 2017; Chartzoulakis and Bertaki, 2015; Bongiovanni and Lowenberg-Deboer, 2004), such as Goal 2 (Zero Hunger) and Goal 15 (Life on Land) of the United Nations Sustainable Development Goals (SDGs) (Hou et al., 2020; UN, 2021). Generally, management zones aim to consider the impact of various factors that can influence crop productivity and result in yield gaps, a key one being soil heterogeneity and health (Licker et al., 2010). Soil systems can be relatively static in time (Arshad et al., 2015) and are fundamental due to their multifunctional role and impact on ecosystem services (Hamidov et al., 2018). Within these systems, soil properties such as texture, organic matter content, cation exchange capacity, and bulk density greatly influence soil moisture dynamics, salinity, nutrient availability, and other variables affecting crop yield (Kibblewhite et al., 2008; Dobarco et al., 2021) and are thus a good target for management zone delineation. However, soil heterogeneity is not solely responsible for yield losses, and effective management zones should also incorporate other influencing factors to provide a comprehensive and holistic management solution.

Traditional methods for soil characterization to support management zone delineation (Brogi et al., 2021; Geologischer Dienst NRW) generally rely on laborious in-situ sampling and laboratory analysis, which may fail in capturing soil variability with sufficient detail (Kuang et al., 2012). In

recent years, advances in proximal soil sensing, defined as methods that utilize sensors positioned
near or in direct contact with the soil (Adamchuk et al., 2017), have provided valid alternatives to
direct soil sampling (Pradipta et al., 2022). In particular, non-invasive agro-geophysical methods
such as electromagnetic induction (EMI) have proven suitable for management zone delineation
due to their high mobility (Binley et al., 2015; Garré et al., 2021) and the fact that the measured
apparent electrical conductivity (ECa) of the soil is related to key soil properties, such as soil
salinity, soil water content, texture, compaction, and organic matter content (Corwin and Lesch,
2003; Abdu et al., 2008; Altdorff et al., 2017; Jadoon et al., 2015; Robinet et al., 2018; Zhu et al.,
2010; von Hebel et al., 2018). Modern EMI devices are able to efficiently provide soil information
for multiple depth ranges thanks to multi-coil instrumentation (Rudolph et al., 2015; von Hebel et
al., 2014; Blanchy et al., 2024; Lueck and Ruehlmann, 2013; Corwin and Scudiero, 2019),
especially when supported by a moderate amount of ground truth data (Brogi et al., 2019).
However, the use of EMI alone can show limitations in capturing local aspects that have an impact
on yield but that are not strongly influenced by soil variability. For instance, pest and weed
infestations can drastically reduce crop productivity, and these factors may not correlate directly
with soil variability (Becker et al., 2022; López-Granados, 2011). Additionally, climate change
impacts, such as altered precipitation patterns and temperature fluctuations, can affect crop health
and yield in ways that EMI cannot detect (Pradipta et al., 2022). Finally, it is also important to
stress that accurate EMI mapping generally requires optimal conditions like bare soil, favourable
weather, and absence of confounding factors (James et al., 2003).

An alternative to proximal soil sensing for the delineation of management zones is the use of
remote sensing approaches, which enables efficient large-scale data acquisition without the need

for direct physical access to the investigated area (Weiss et al., 2020). By using sensors mounted on satellites, airplanes, or drones, remote sensing monitors parameters related to crop health and development (Jin et al., 2019; Liaghat and Balasundram, 2010). For example, vegetation indices such as the Normalized Difference Vegetation Index (NDVI) are generally well-established, simple, and effective proxies for crop health (Carfagna and Gallego, 2005; Stamford et al., 2023; Wang et al., 2020; Xue and Su, 2017). High-resolution (<5 m) data products from satellites are being increasingly used in precision agriculture (Mohammed et al., 2020; Trivedi et al., 2023). Also, remote sensing platforms like PlanetScope, Sentinel-2, and Landsat offer frequent revisit times, thus providing sufficient temporal resolution to track changes in plant health throughout the growing season (Hunt et al., 2019; Skakun et al., 2021). Despite these advantages, remote sensing data are affected by cloud cover or other sub-optimal meteorological conditions (Wilhelm et al., 2000) and primarily capture above-ground information on plant health and biomass, and can thus struggle to provide direct information about the interplay between soil conditions and crop development.

Several studies have explored a combination of EMI and remote sensing methods for the delineation of management zones. For example, von Hebel et al. (2021) combined EMI and drone-based NDVI measurements and found that EMI-based management zones offered consistent insights into soil texture and water content, while the added value of NDVI strongly depended on the timing of the drone measurements and thus on the specific crop conditions. In a similar study, Esteves et al. (2022) showed that integration of EMI and NDVI from Sentinel-2 (10 m resolution) effectively provided zones with distinct soil and crop nutrient characteristics. However, they reported a negative relationship between ECa and NDVI due to local magnesium imbalances and

vegetation stress. In addition to EMI and remote sensing, historical yield maps can help in
identifying yield trends across years and different cultivated crops. For example, Ali et al. (2022)
integrated seven years of yield data with Landsat-based NDVI and soil sampling over a 9 ha field,
but ultimately could obtain only a limited subdivision of the field into two management zones with
a relatively low resolution of 30 m. Overall, previous studies have made important contributions
towards integrating EMI and NDVI data for improved management zone delineation (Corwin and
Scudiero, 2019; Ciampalini et al., 2015). However, the results can be influenced by data resolution
and acquisition timing as well as by local management and soil-plant interactions, with some
studies suggesting that EMI alone can offer sufficient insights into soil patterns (Esteves et al.,
2022; von Hebel et al., 2021). Nonetheless, the added value of NDVI holds unexplored potential
due to the higher spatial and temporal resolution of recent satellite platforms (Breunig et al., 2020;
Georgi et al., 2018).

Machine learning clustering algorithms have been widely used to delineate management zones
from spatially distributed datasets such as EMI or NDVI (Saifuzzaman et al., 2019; Castrignanò
et al., 2018; Chlingaryan et al., 2018; Zhang and Wang, 2023). For example, Wang et al. (2021)
used supervised Random Forest classification for combining EMI data with environmental
covariates to predict soil salinity. Similarly, Brogi et al. (2019) employed supervised learning to
combine EMI with soil sampling and generate high-resolution soil maps for a 1 km² agricultural
area. However, the results of supervised classification approaches may depend on the interpreter
and often need expert knowledge as well as extensive ground-truth data for training (Liakos et al.,
2018; Usama et al., 2019). K-means and ISODATA clustering are unsupervised methods used to
delineate management zones (Bijeesh and Narasimhamurthy, 2020; Ylagan et al., 2022; Tagarakis
et al., 2013) but these approaches can be sensitive to initial conditions and struggle to handle non-
linear relationships in datasets (Geng et al., 2020; Li et al., 2018). Thus, more advanced methods
such as self-organizing maps (SOM) have been successfully used to analyse complicated data
structures provided by proximal and remote sensing data (Romero-Ruiz et al., 2024; Moshou et
al., 2006; Taşdemir et al., 2012). A remaining key challenge of unsupervised methods is the
definition of the optimal number of clusters. Widely used approaches such as the elbow and
silhouette method (Saputra et al., 2020) often struggle when applied to non-linearly distributed or
spatially complex datasets (Schubert, 2023), and may thus require subjective judgment or expert
knowledge (Liang et al., 2012). To address this challenge, the Multi-Cluster Average Standard
Deviation (MCASD) approach that relies on an evaluation of the intra-cluster variability has
recently been introduced (O'Leary et al., 2023) and successfully applied to the integration of
complex spatial datasets (O'Leary et al., 2024). However, many of these novel approaches have
seen limited applications in agricultural contexts (Khan et al., 2021) and the added value of
delineating management zones from datasets of different origin remains unaddressed (Koganti et
al., 2024).

Within this context, this study expands on previous research by combining high- resolution multi-
coil EMI and satellite-based NDVI data within a harmonized framework, applying consistent
normalization, and validating the resulting zones with multi-year yield data and dense soil
sampling. The potential of delineating management zones by integrating EMI and NDVI is
explored for a single 70 ha agricultural field near Berlin, Germany. Management zones were
derived using three data sources: i) ECa maps from nine different depths of investigation (DOI)
obtained with EMI between 2022 and 2024, ii) seven NDVI images obtained from PlanetScope in
2019, and iii) a combination of EMI and NDVI data. Management zones were delineated using
SOM while the optimal number of clusters was obtained with the MCASD method. In a following
step, the number of clusters was refined using post-hoc analysis using a large dataset of soil
samples and yield maps at 10 m resolution from 2011 to 2019. Finally, it was evaluated to what
extent management zones derived from EMI, NDVI, or a combination of both represent soil
characteristics and yield patterns using visual inspection and statistical analysis.

**2 Materials and Methods**
**2.1 Study area**
The study site is part of the patchCROP (patchCROP, 2020) landscape laboratory of the Leibniz
Centre for Agricultural Landscape Research (ZALF) near Tempelberg, Brandenburg, Germany
(52.4426 N, 14.1607 E, altitude 68 m). It is located in the transition zone between humid oceanic
and dry continental climate. The long term average temperature from 1980 to 2020 was 8.3°C and
the mean annual precipitation for the same period was 533 mm (DWD, 2021; Koch et al., 2023).
The investigated field has an area of approximately 70 ha (Figure 1). Until 2020, this field was
managed as a single unit. In March 2020, the patchCROP experiment was established to study the
impact of landscape diversification through the use of smaller field sizes, site-specific crop
rotations, different field management practices, and the use of new technologies including
proximal soil sensing, remote sensing, and robotic technologies (Grahmann et al., 2021). For this,
thirty patches of 72 x 72 m were established within the investigated field (Donat et al., 2022) (Fig.
1). In terms of geomorphology, the site is described as a young moraine landscape shaped by past
glaciations, and characterized by an undulating relief and heterogeneous soil characteristics (Koch
et al., 2023; Öttl et al., 2021; Meyer et al., 2019). The topsoil is predominantly sandy, but a more
clayey layer is present at different depths in the subsoil (Hernández-Ochoa et al., 2024).

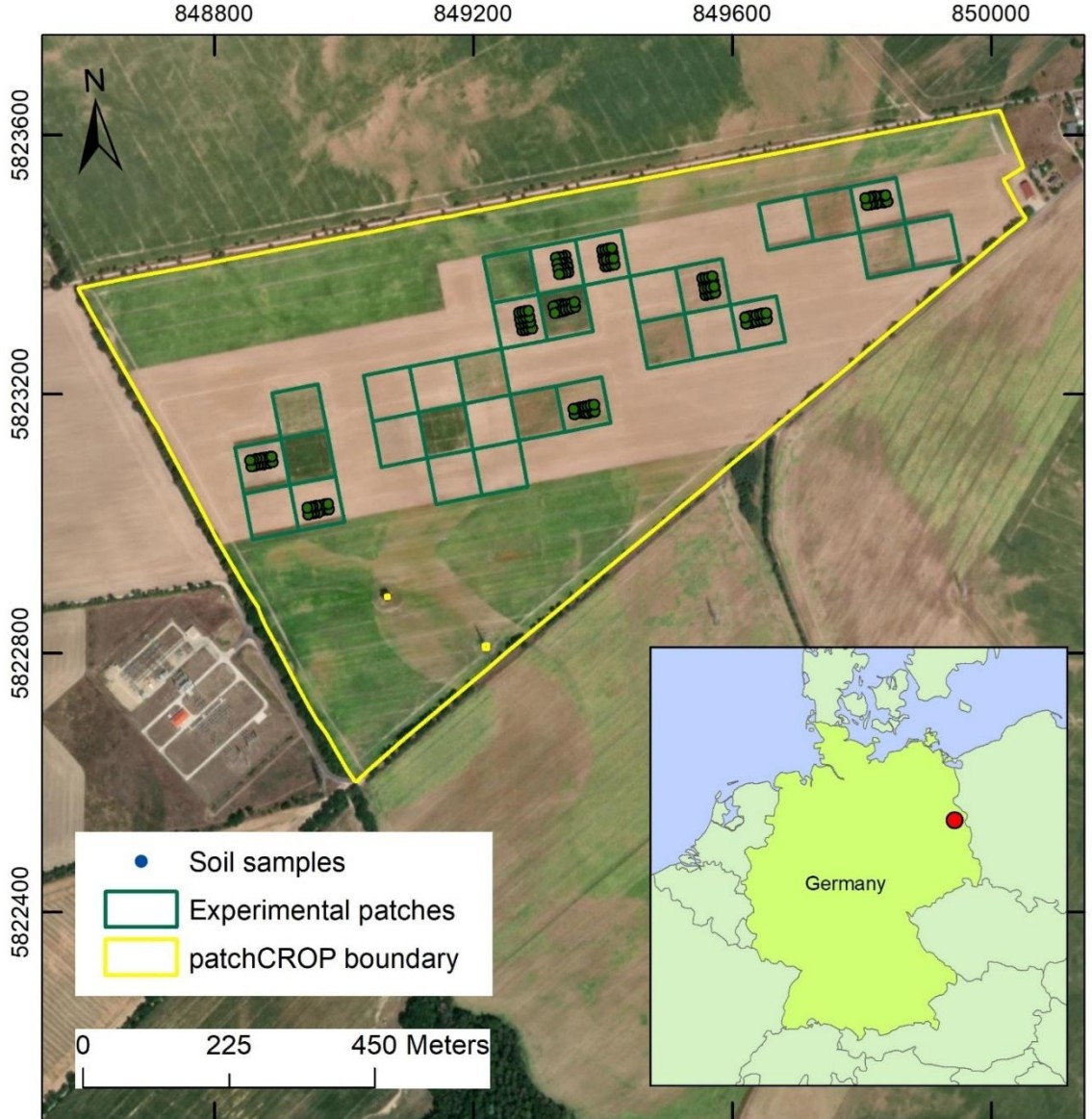


Figure 1. Overview of the patchCROP study area in Tempelberg, Brandenburg (ESRI, 2020). The
yellow border indicates the boundary of the investigated field, whereas the green boxes indicate
the thirty patches of the patchCROP landscape experiment. The inset map shows the location of
the study site within Germany; the red dot indicates the site location in Tempelberg.

## 2.2 Data collection and processing

The overall methodology of this study is summarized in Figure 2. This flowchart highlights the

role of EMI and NDVI datasets in the clustering process and the use of multi-year yield maps and

soil samples for validation and refinement of the resulting management zones. More details are

provided in the subsequent sections.





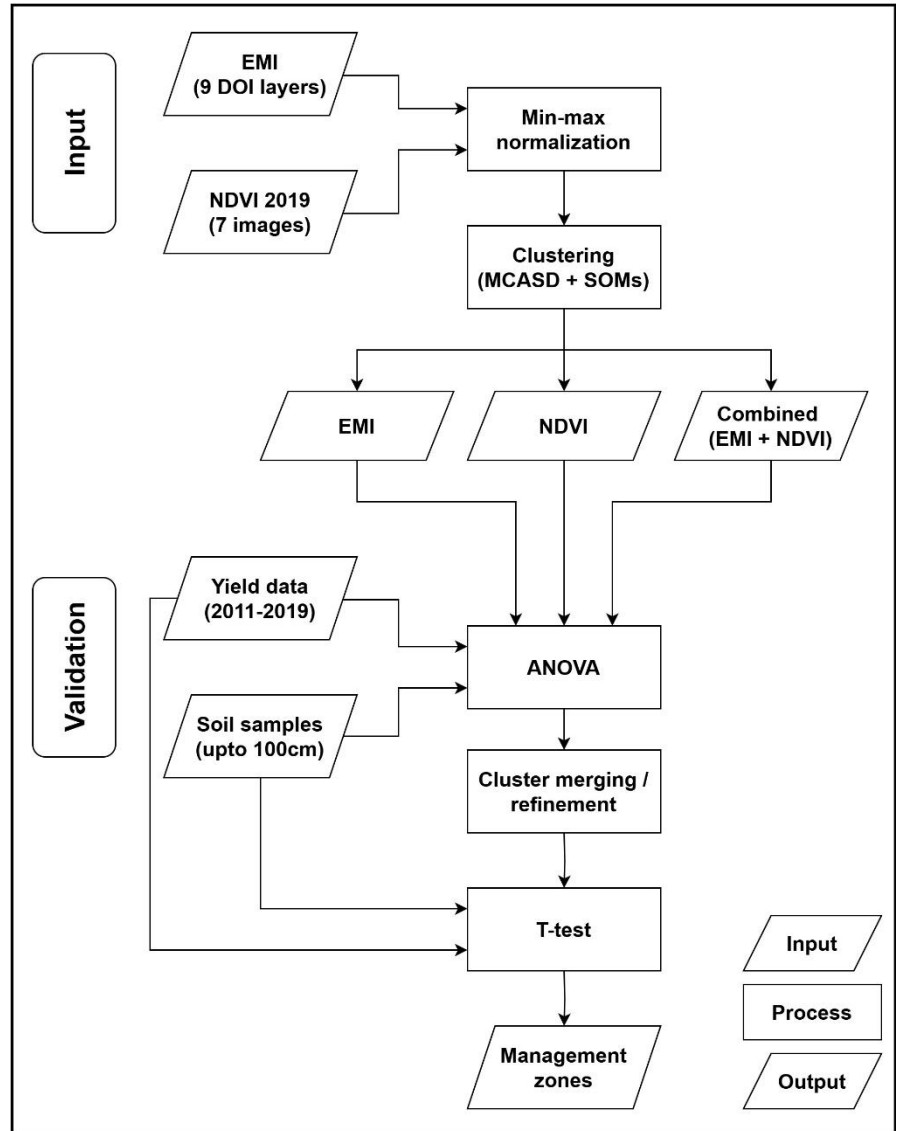


Figure 2. Workflow diagram showing the integration of proximal (EMI) and remote sensing (NDVI) data for unsupervised clustering using MCASD and SOMs. Yield and soil datasets were used for post-hoc validation and refinement of management zones.

**2.2.1 Electromagnetic Induction (EMI) measurements**

Frequency-domain EMI devices generate a fixed-frequency alternating current in a transmitter coil, which produces a primary magnetic field. This primary magnetic field induces eddy currents in the soil, thus generating a secondary magnetic field. The primary and secondary magnetic fields

are sensed by a receiver coil. The quadrature component of the ratio between the primary and
secondary magnetic fields is directly proportional to the apparent electrical conductivity (ECa) of
the ground (Keller and Frischknecht, 1966; Ward and Hohmann, 1988; McNeill, 1980). The
measured ECa is strongly affected by soil properties such as salinity, water content, clay content
(and thus texture), compaction, and to a lesser degree organic matter content and cation exchange
capacity (Corwin and Lesch, 2005; Robinet et al., 2018). The depth sensitivity of EMI
measurements depends on coil spacing and coil orientation. Larger spacing results in increased
depths of investigation (DOI), while the coil orientation influences the sensitivity to shallow or
deep subsurface (Lavoué et al., 2010; Simpson et al., 2009).

In this study, two EMI devices were used simultaneously: a CMD-Mini Explorer (GF Instruments,
Brno, Czech Republic) with three receiver coils oriented in a vertical coplanar configuration
(VCP), and a custom-made CMD-Mini Explorer – Special Edition equipped with six receiver coils
oriented in a horizontal coplanar configuration (HCP). The VCP configuration is most sensitive to
the shallow subsurface, with decreasing sensitivity as depth increases. In contrast, the HCP
configuration is less sensitive to the shallow subsurface, with sensitivity peaking at a depth of
approximately 0.4 times the coil separation (McNeill, 1980). As a rule of thumb, the DOI for the
VCP setup is approximately 0.75 times the coil separation. For the HCP setup, the DOI is
approximately 1.5 times the coil separation. For the set-up used here, the resulting DOI ranges
from 0-24 to 0-270 cm. Details of the EMI set-up are summarized in Table 1.

Table 1. Details of the two EMI devices with coil number, orientation, separation, DOI, and frequency.

| EMI device | Receivers | Orientation | Separation (cm) | DOI (cm) | Frequency (Hz) |
|---|---|---|---|---|---|
| Mini Explorer | 3 | VCP | 32 | 0-24 | 30 |
| | | VCP | 71 | 0-53 | |
| | | VCP | 118 | 0-89 | |
| Mini Explorer | 6 | HCP | 35 | 0-52 | 25.17 |
| Special Edition | | HCP | 50 | 0-75 | |
| | | HCP | 71 | 0-108 | |
| | | HCP | 97 | 0-146 | |
| | | HCP | 135 | 0-203 | |
| | | HCP | 180 | 0-270 | |

Due to the ongoing PatchCROP experiment with small patches using variable cropping systems, it was not possible to cover the entire field in a single EMI campaign. EMI data were thus collected in four campaigns conducted between August 2022 and October 2024. During each campaign, the EMI devices were placed in sleds and warmed up for approximately 30 minutes before use. The sleds were then pulled by an all-terrain vehicle (ATV) at a speed of approximately 6 to 8 km/h. Data collection occurred at a frequency of 0.2 s, resulting in an inline spatial resolution of 0.25 to 0.50 m. A track spacing of ~2.5 m was used within the experimental patches and a track spacing between 5 to 45 m (typically well below 10 m) was used in the rest of the field. A Real Time eXtended (RTX) center point differential global positioning system (DGPS) (Trimble Inc., Sunnyvale, United States) was used to record the position of the sleds with centimeter accuracy.

For more information about the setup for EMI measurements, the reader is referred to von Hebel
et al. (2018).

The measured ECa values were filtered using a Python-based method similar to the approach of
von Hebel et al., (2014), which has been successfully applied in several studies (Brogi et al., 2019;
von Hebel et al., 2021; Kaufmann et al., 2020; Schmäck et al., 2022). The first filter removes
values that are deemed too high or too low based on user-defined thresholds (-50 and 50 mS/m in
this study). A second filter divides the data into a user-defined number of bins (20 in this study)
and removes the data from bins with a low fraction of measurements (<1% in this study). In a third
step, a spatial filter is used to identify and discard ECa values that deviate from adjacent positions
more than a given amount (1 mS/m in this study) to avoid unrealistically high lateral ECa
variations. After the application of these three filters, ~5% of the measured ECa values were
removed, although this value varied between measurement campaigns.

Given that the EMI data were acquired in four campaigns with different environmental conditions
(e.g. soil water content, soil temperature), each EMI acquisition campaign was separately
normalized by using a standardized z-score normalization method as used by Rudolph et al. (2015):

$ECa_{z,i} = (ECa_i - \mu_i)/\sigma_i$ (1)

where $ECa_{z,i}$ is the normalized $ECa$ value for the i-th campaign, $ECa_i$ is the measured ECa value
for the i-th campaign, $\mu_i$ is the mean ECa value of the i-th campaign, and $\sigma_i$ is standard deviation
of ECa values for the i-th campaign. Following normalization, manual cleaning was conducted in
ArcMap v10.8.2 (ESRI, Redlands CA, USA) to remove points typically occurring at the start and
end of each campaign or in short periods where the EMI system was left stationary. In the final
step, the normalized data for each of the nine coil configurations were interpolated to a regular 3
by 3 m grid using ordinary Kriging with a gaussian semivariogram and merged into a single
multidimensional raster mosaic dataset.

**2.2.2 Remotely sensed NDVI data**
High-resolution PlanetScope Level 3B satellite images from the 2019 growing season (winter rye)
were used to obtain NDVI maps. Between 01-Jan-2019 and 31-July-2019, 48 cloud free images
were available. Seven of these images were selected to represent crop development during the
growing season. PlanetScope image products are pre-processed and have already undergone
radiometric and atmospheric corrections. No additional pre-processing was required. The
PlanetScope sensor captures spectral information in four bands: blue (B1), green (B2), red (B3),
and near-infrared (NIR - B4) with a spatial resolution of 3 m. The normalized difference vegetation
index (*NDVI*) was calculated using the reflectance in the red (*R*) and near-infrared bands (*NIR*):

$NDVI = (NIR - R)/(NIR + R)$ (2)

The resulting NDVI values range from -1 to 1, where values close to 1 indicate healthy vegetation,
and values close to zero or negative values generally represent non-vegetated surfaces, senescent,
stressed or unhealthy plants or dry vegetation, or features such as clouds and water that exhibit
lower NIR reflectance (Sishodia et al., 2020).

**2.2.3 Yield data**

Georeferenced yield maps of nine growing seasons (2011-2019) were used. These yield maps were generated using a yield monitoring system (CLAAS Quantimeter, Hersewinkel, Germany) mounted on two different combine harvesters. From 2011 to 2013, data were collected using a CLAAS 580. From 2014 onwards, a CLAAS Lexion 770 TT was used. In the 2011 – 2019 period, the field was either cultivated with winter rye (2011, 2013, 2014, 2016, 2017, and 2019) or rapeseed (2012, 2015, and 2018). For additional details on data processing and yield map generation, readers are referred to Donat et al. (2022). The original yield data from Donat et al. (2022) were available as georeferenced yield data points with a spacing of ~10 m. These points were interpolated to a regular grid with 10 m resolution using ordinary kriging.

**2.2.4 Soil sampling and data on soil characteristics**

Extensive soil sampling campaigns were conducted between 2020 and 2024, focusing on the experimental patches within the 70 ha field. At 160 locations, soil samples up to 100 cm depth were obtained using a Pürckhauer soil auger with an 18 mm inner diameter. The soil properties analyzed in this study included the depth of soil texture transition, defined as the depth (in cm) at which the sandy top layer ends (EOS (end of sandy layer) in the following), as well as the soil texture (percentage of sand, silt, and clay) of the top sandy layer and the layer below. Soil texture was determined by using the wet sieving and sedimentation method (ISO, 2002). The particle size distribution was defined according to the IUSS Working Group 150 WRB guidelines (IUSS Working Group, 2015). When multiple subsamples for a single layer were available at a given location, weighted averages of sand, silt, and clay fraction for the whole layer were obtained using the thickness of each subsample.

310

**2.3 Clustering for delineation of management zones**

Three different data combinations were created and investigated: a) EMI maps, b) time-series of NDVI maps, and c) a combination of the EMI maps and NDVI maps. Before clustering, a standard preprocessing step of normalization was applied on each dataset to ensure that variables with different ranges and units contribute equally in the classification process. The choice of normalization method can be particularly important when combining datasets with different scales, such as EMI and NDVI, to prevent dominance of one dataset over the other and to maintain the integrity of the input features In this study, a min-max scaling was applied, where all values were rescaled to a standard range between 0 and 1 (Patro and Sahu, 2015). For EMI, a single normalization was applied to the nine $ECa_z$ maps. In this case, the min-max normalization used the minimum ($ECa_{z\,min}$) and maximum value ($ECa_{z\,max}$) from all nine 9 maps:

$$ECa_z' = \frac{ECa_z - ECa_{z\,min}}{ECa_{z\,max} - ECa_{z\,min}} \tag{3}$$

where $ECa_z$ is the original value, and $ECa_z'$ is the normalized value. For NDVI, each of the seven NDVI maps was normalized independently:

$$NDVI'_i = \frac{NDVI_i - NDVI_{i,min}}{NDVI_{i,max} - NDVI_{i,min}} \tag{4}$$

where $NDVI'_i$ is the normalized value for the i-th map, $NDVI_i$ is the original value of NDVI of the i-th map, $NDVI_{i,\,min}$ and $NDVI_{i,\,max}$ are the minimum and maximum values of the i-th NDVI map. This difference in normalization was necessary to preserve the depth-dependent structure of EMI

data, as ECa represents a bulk measurement where each reading is influenced by adjacent depths.
In contrast, NDVI measurements are independent and acquired at different time points, and thus
reflect temporal variations in vegetation dynamics.

In this study, a Self-Organizing Map (SOM), an unsupervised machine learning classification
technique, was used for clustering (Kohonen, 2013). SOM is a centroid-based clustering technique,
similar in some aspects to K-means clustering (Celebi et al., 2013). While K-means clustering
assigns each data point to a cluster based on the minimum distance to the cluster centroid in the
data space, SOM utilizes an artificial neural network to organize and visualize high-dimensional
data (Valentine and Kalnins, 2016). The key distinction lies in how SOM projects the data onto a
two-dimensional grid while preserving the topological relationships of the input data. Each data
vector in SOM is assigned to a numerical cluster, where the cluster centre is representative of all
the data points associated with it. These cluster centres, which have dimensions similar to the input
data vectors, adjust iteratively during the training process to better represent the underlying data
distribution. This approach allows SOM to effectively map complex data patterns while
maintaining the spatial relationships between clusters.

The Multi-Cluster Average Standard Deviation (MCASD) approach was used to determine the
optimal number of clusters for SOM. This method evaluates the stability of the cluster centres in
the dataspace over multiple clustering attempts as the number of clusters increases. This metric
assumes that an appropriate number of clusters for a dataset is any at which the cluster centres do
not vary significantly when the clustering algorithm is run multiple times. In this study, MCASD
analysis was tested with a maximum number of 20 clusters with 100 SOM clustering runs for each
number of clusters to calculate the MCASD stability metric. The number of clustering runs was
determined during preliminary testing, where it was observed that most datasets stabilized in terms
of variability between 70 and 80 iterations. To ensure consistency and reproducibility, we adopted
100 runs per cluster number. Upon completion of MCASD analysis, the highest number of clusters
with a low MCASD metric was selected, as this represents the maximum resolution of the spatial
variability that can be obtained through clustering (O'Leary et al., 2023). This clustering process
was performed in MATLAB v2023a (MathWorks, Natick, Massachusetts, USA).

**2.4 Statistical analysis**
To assess the differences between clusters derived from the three datasets, a one-way analysis of
variance (ANOVA) was conducted in SPSS (IBM, Chicago, IL, United States). ANOVA was used
to identify whether there were significant differences between clusters in terms of soil properties
or yield using a significance threshold of $p < 0.05$. Following the ANOVA, a Tukey's HSD
(Honestly Significant Difference) test was used as a post-hoc analysis to determine which of the
clusters were significantly different. In this step, the depth of the sandy layer, the texture of the
overlying layer, the texture of the layer below, and the yield data were used. Thus, this step is
complimentary to the previous cluster selection step with MCASD, which did not consider soil
and yield data. Clusters that did not exhibit significant differences were merged during a
reclassification step, refining the clustering results to ensure that each final cluster was distinct and
statistically meaningful, both in terms of the input datasets and in terms of soil properties and yield.
The latter was confirmed using two tailed t-tests between matching layers of adjacent soil classes
in the reclassified map.

## 3 Results and Discussion

### 3.1 ECa$_z$, NDVI, and yield maps

The ECa$_z$, NDVI, and yield maps highlight unique aspects of field heterogeneity and offer insights into subsurface soil properties, above-ground crop performance, and their combined effects on productivity. In the following, these input datasets for management zone delineation are briefly introduced.

### 3.1.1 EMI maps

Nine ECa maps with 3 m resolution were obtained from the interpolation of the nine coil configurations recorded during the EMI measurements. The results for one coil configuration (HCP 050 cm) are exemplary shown in (Figure 3) before and after normalization. The study area was measured under varying conditions in terms of soil temperature, soil moisture, and effect of agricultural management. This resulted in differences of average ECa and spatial patterns (Figure 3a). Although it is well known that temperature affects measured ECa (Pedrera-Parrilla et al., 2016; Vogel et al., 2019), it was not possible to perform a comprehensive temperature correction in this study due to the lack of sufficient soil temperature data. Moreover, it has been shown that temperature correction has limitations compared to normalization methods when the dataset is composed of various depths of investigation and is affected by multiple agricultural management practices (Brogi et al., 2019; Rudolph et al., 2015). Thus, Z-score normalization was applied for each measurement campaign to reduce the differences between data measured on different days. Figure 3b shows the normalized EMI map for the same coil configuration as shown in (Figure 3a). The normalization successfully harmonized the data, minimizing the influence of varying soil moisture and temperature during acquisition, resulting in more consistent spatial patterns that

better represent subsurface soil properties. However, some localized artefacts in the normalized
maps still persist. For example, areas near the field boundaries or experimental patches exhibit
subtle inconsistencies that may be influenced by edge effects or localized disturbances. Despite
these minor limitations, the normalized ECa maps provide a robust foundation for further analysis
and management zone delineation.

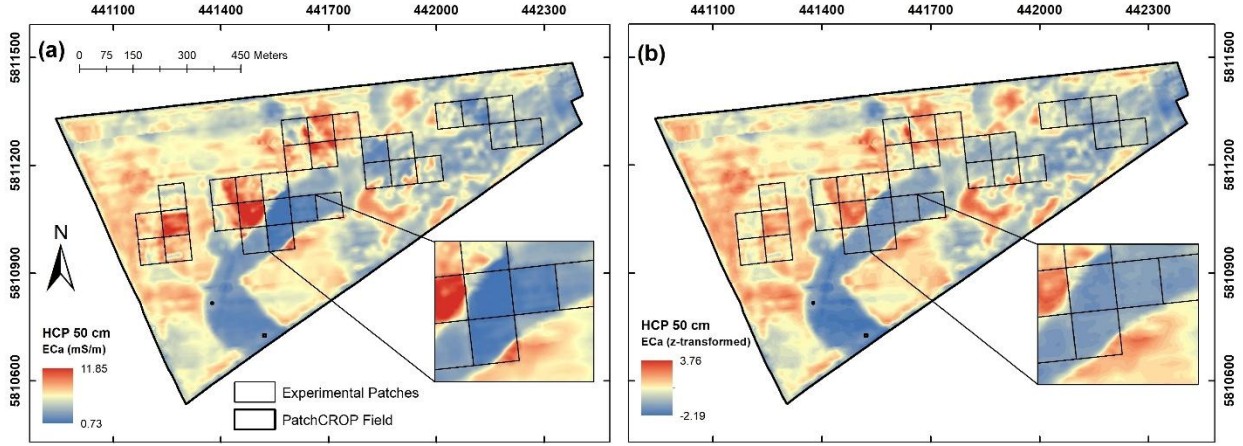


Figure 3. Comparison of apparent electrical conductivity (ECa) maps before and after z-score
normalization for the HCP 50 cm configuration with **(a)** the non-normalized ECa map, where the
zoomed-in section highlights the influence of varying environmental conditions such as soil
moisture and temperature leading to inconsistent patterns and **(b)** the z-score normalized ECa map,
which minimizes the influence of these external factors.

Figure 4 shows the nine normalized $ECa_z$ maps for the VCP and HCP configurations. These maps
display heterogeneous patterns of ECa, primarily attributed to variations in soil characteristics in
space and with depth. A prominent feature is the elongated channel extending from the northeast
to the southwest of the field, which represents areas with lower $ECa_z$ values. This feature is
associated with sandy soils that generally hold less water and nutrients, indicating a coarse-
textured zone with lower electrical conductivity. In contrast, the northwest and southeast regions
of the field exhibit medium to high $ECa_z$ values, which may reflect areas of higher moisture content
and finer soil particles, such as loamy textures. Additionally, in the northeastern part of the field,
a more heterogeneous area with short-scale variations can be observed where the $ECa_z$ values vary
considerably between the nine maps. For the shallow VCP configurations, this area shows low
$ECa_z$ values, which are indicative of sandy soils or dry conditions near the surface. For the deeper
HCP configurations, this same area shows higher $ECa_z$ values, suggesting an increase in soil
moisture or finer soil texture at greater depths. This pattern highlights the layered soil
heterogeneity in this region, with subsurface properties differing significantly from the surface.
Overall, the EMI data reveal a high degree of spatial variability and provide valuable insights into
subsurface soil variability, which is critical for precision agricultural management.

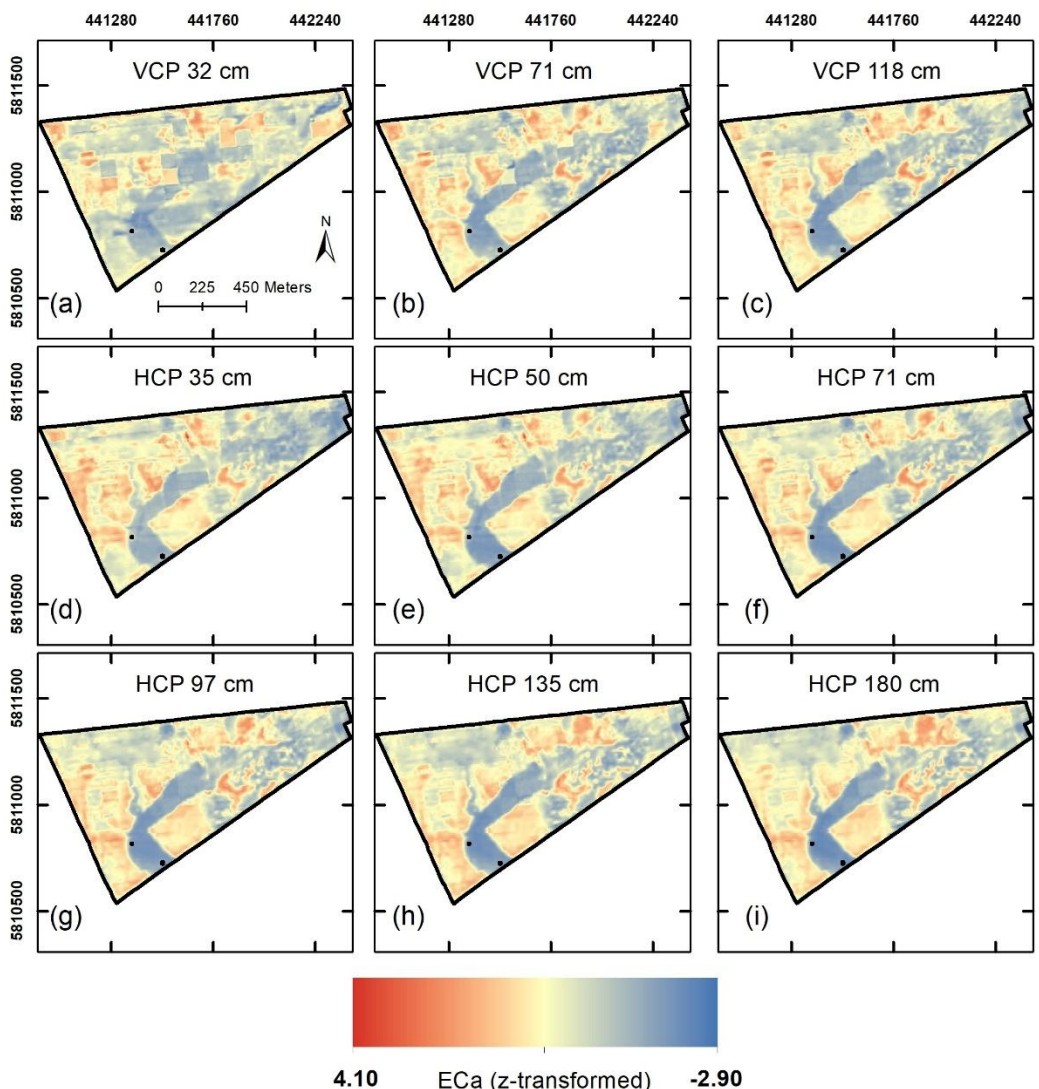


Figure 4. Normalized apparent electrical conductivity (ECaz) maps derived from electromagnetic

induction (EMI) measurements using multiple coil separations in vertical coplanar (VCP) and

horizontal coplanar (HCP) configurations (see Table *1* for more details). These maps highlight the

spatial variability of subsurface soil properties, with higher ECaz values (red) indicating areas of

higher moisture retention or finer soil textures, and lower ECaz values (blue) corresponding to

sandy soils with lower conductivity.

439

440

### 3.1.2 NDVI maps

All available PlanetScope satellite images for the growing season 2019 (winter rye) were visually evaluated to assess their usability. Before April 2019, no meaningful patterns in NDVI were observed due to the relatively short height (10 to 20 cm) and low biomass of winter rye and the lack of water- or nutrient-induced stress in this early growth stage. Moreover, images from July 2019 were excluded from the analysis as the crop had reached maturity, and no further growth or development was evident. By this time, the physiological activity of the plants had ceased, and harvesting was completed on 04/08/2019.

After this initial analysis, seven NDVI images spanning the period between April and June, hence from flowering to maturity, were selected for further analysis (Figure 5). The descriptive statistics of the NDVI data are given in Table 2 and show a high degree of temporal variation. Following crop development during the growing season, the mean NDVI peaked on 30 April 2019 (221 days after sowing). Afterwards, NDVI values gradually declined as the crop approached maturity, which is consistent with physiological changes during growth of winter rye (Hatfield and Prueger, 2010). Figure 5 illustrates the temporal development of the spatial variation of NDVI, highlighting the spatial heterogeneity of crop performance within the field (especially Figure 5d-g) where areas of lower NDVI are associated with poorer crop performance and areas of higher NDVI indicate healthier crops. Generally, the key patterns in crop performance are in good agreement with the patterns observed in the EMI maps. Areas with persistently low NDVI values generally correspond to areas with low $ECa_z$, and areas with high NDVI values mostly correspond to areas with high $ECa_z$. However, differences between patterns in NDVI and EMI can also be found. This is expected given that the dynamic changes in crop vigour and vegetation health shown by NDVI are not solely

related to subsurface soil conditions captured by EMI. For example, specific areas with low NDVI values were observed in regions of medium to high $ECa_z$, possibly reflecting localized crop stress due to non-soil-related factors such as disease, waterlogging, or nutrient imbalances.

Table 2. Summary of remotely sensed NDVI imagery and corresponding dates after sowing.

| Date of acquisition | Days after sowing | Mean NDVI | Max NDVI | Min NDVI |
|---|---|---|---|---|
| 05 April 2019 | 196 | 0.67 | 0.78 | 0.42 |
| 16 April 2019 | 207 | 0.72 | 0.85 | 0.46 |
| 30 April 2019 | 221 | 0.76 | 0.88 | 0.38 |
| 11 May 2019 | 232 | 0.61 | 0.71 | 0.34 |
| 30 May 2019 | 251 | 0.58 | 0.66 | 0.41 |
| 12 June 2019 | 263 | 0.49 | 0.65 | 0.31 |
| 24 June 2019 | 276 | 0.49 | 0.71 | 0.30 |

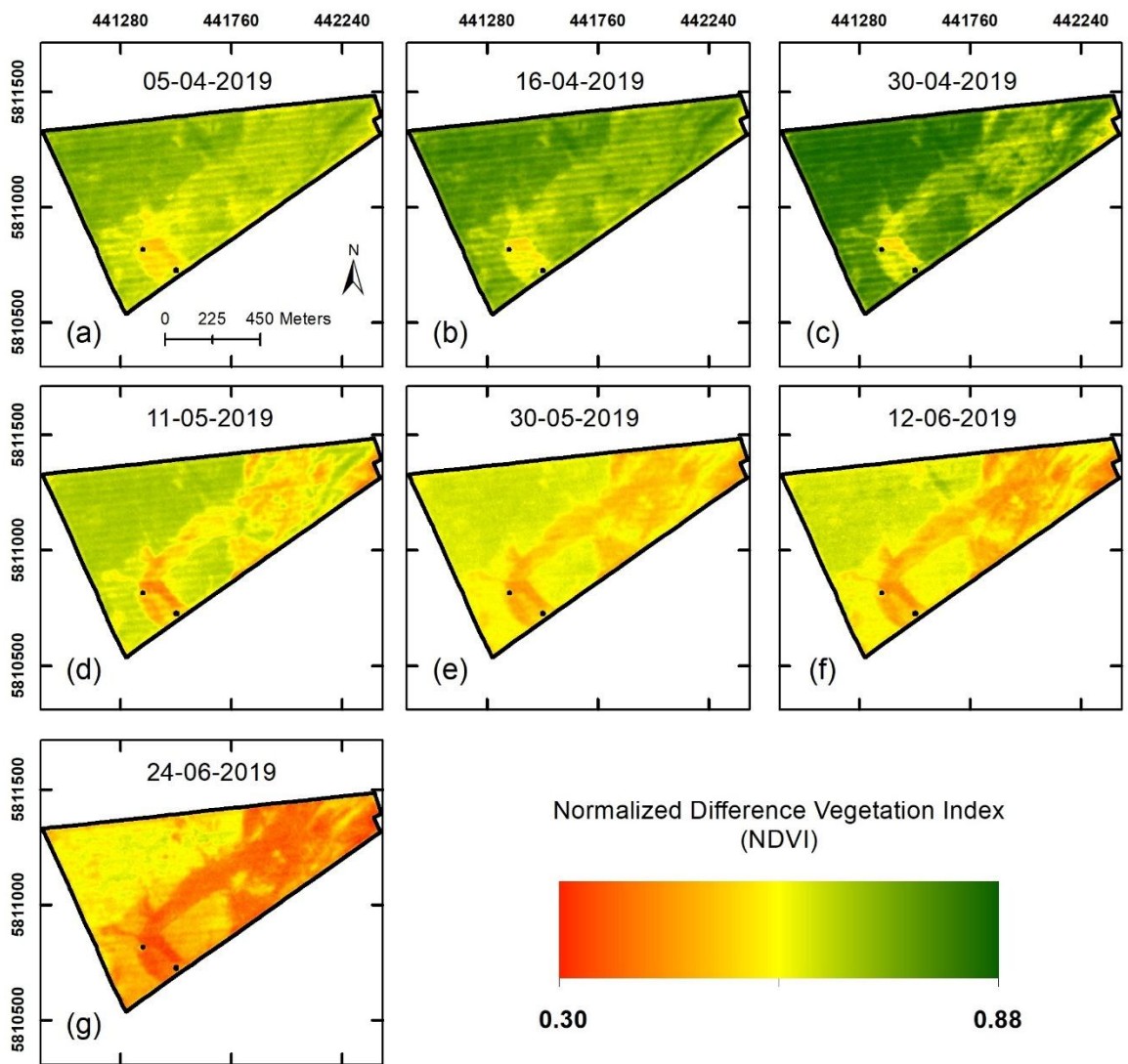

Figure 5. Seven NDVI maps derived from PlanetScope satellite imagery representing the temporal

variability in vegetation development during the 2019 growing season. The images, dated from

05-04-2019 to 24-06-2019, capture critical crop growth stages, including flowering and maturity.

### 3.1.3 Yield maps

Figure 6 presents nine years (2011–2019) of yield maps interpolated at a 10 m resolution to

represent spatial variability across the field. The maps illustrate distinct patterns of high and low

productivity areas. Yield variability is consistent across multiple years, although variations in

measured yield can be observed between years. The years 2012 and 2013 show lower quality yield
data due to incomplete datasets (Donat et al., 2022) caused by equipment issues and environmental
challenges during data collection. Despite these limitations, they were retained for spatial context
as they still exhibited consistent patterns with other years, and the maps successfully captured the
general spatial yield trends and heterogeneity of the field. These years were not weighted
differently during validation, and the potential influence of this lack of weighting was mitigated
by evaluating multi-year trends and conducting year-by-year comparisons (see Section 3.4). The
high and low yield zones align with known intrinsic field characteristics, such as soil texture,
moisture retention, and nutrient availability (Grahmann et al., 2024). These yield patterns will
serve as validation for comparing the management zones derived from EMI and NDVI data, as
both datasets aim to explain the variability in productivity.

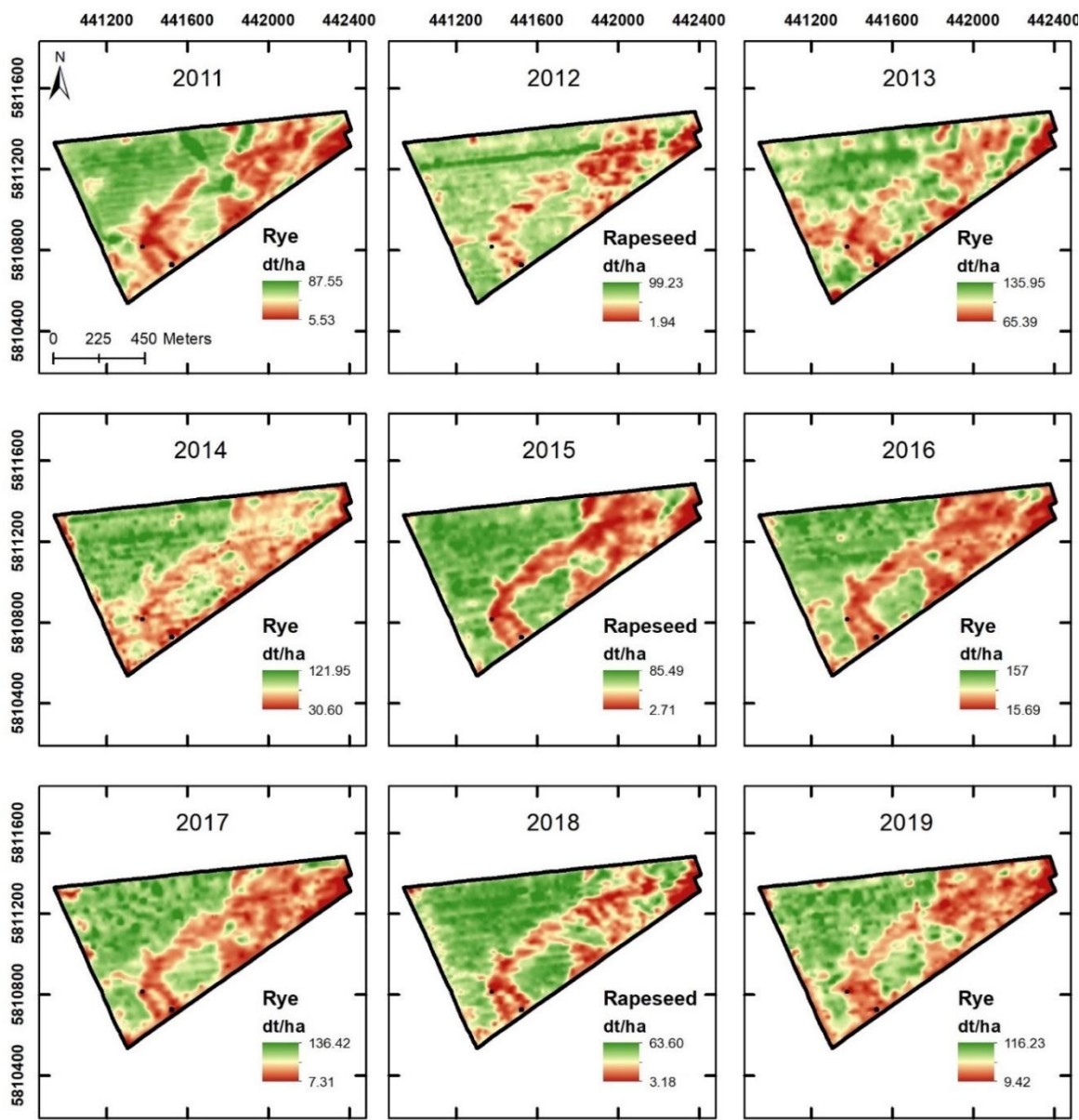


Figure 6: Nine interpolated yield maps (2011–2019) for the patchCROP field showing spatial variability of crop yield at a 10 m resolution. The maps illustrate yield distributions for winter rye (2011, 2013, 2014, 2016, 2017, 2019) and rapeseed (2012, 2015, 2018). High-yield areas (green) and low-yield areas (red) reflect the inherent field heterogeneity. Variability is observed both within and across years, influenced by crop type, management practices, and environmental conditions. The yield range for each year is provided in decitonnes per hectare (dt/ha).

## 3.2 Clustering of EMI and NDVI

The MCASD analysis for the three datasets provided a robust method to determine the optimal number of clusters (Figure 7). The analysis suggested a maximum of five clusters for the EMI data (Figure 7b). These clusters reflect differences in subsurface properties such as soil texture, moisture, and compaction. Cluster 1 corresponds to areas with the highest $ECa_z$ values, which gradually decrease with each subsequent cluster. Cluster 5 represents the lowest $ECa_z$ values. For NDVI (Figure 7e), a maximum of four clusters was selected. While a five-cluster solution was initially identified as viable for NDVI, increasing the number of clusters beyond four did not significantly reduce variability. This made the four-cluster solution more practical and efficient for representing spatial variability in the NDVI data. Cluster 1 identifies areas with relatively high NDVI values, indicative of healthy, dense vegetation and higher crop performance. NDVI values progressively decrease with higher cluster numbers, with cluster 4 showing the lowest values, representing stressed or less productive areas. The combined EMI and NDVI dataset resulted in four clusters (Figure 7h). Visual inspection suggests that both the EMI- and NDVI-based patterns are preserved in the combined dataset, likely due to the min-max scaling applied to standardize each dataset before MCASD analysis (see Appendix A). Clusters 1 and 2 represent areas with high values for both $ECa_z$ and NDVI, while cluster 4 identifies zones with low values for both variables, integrating both above-ground and subsurface variability effectively.

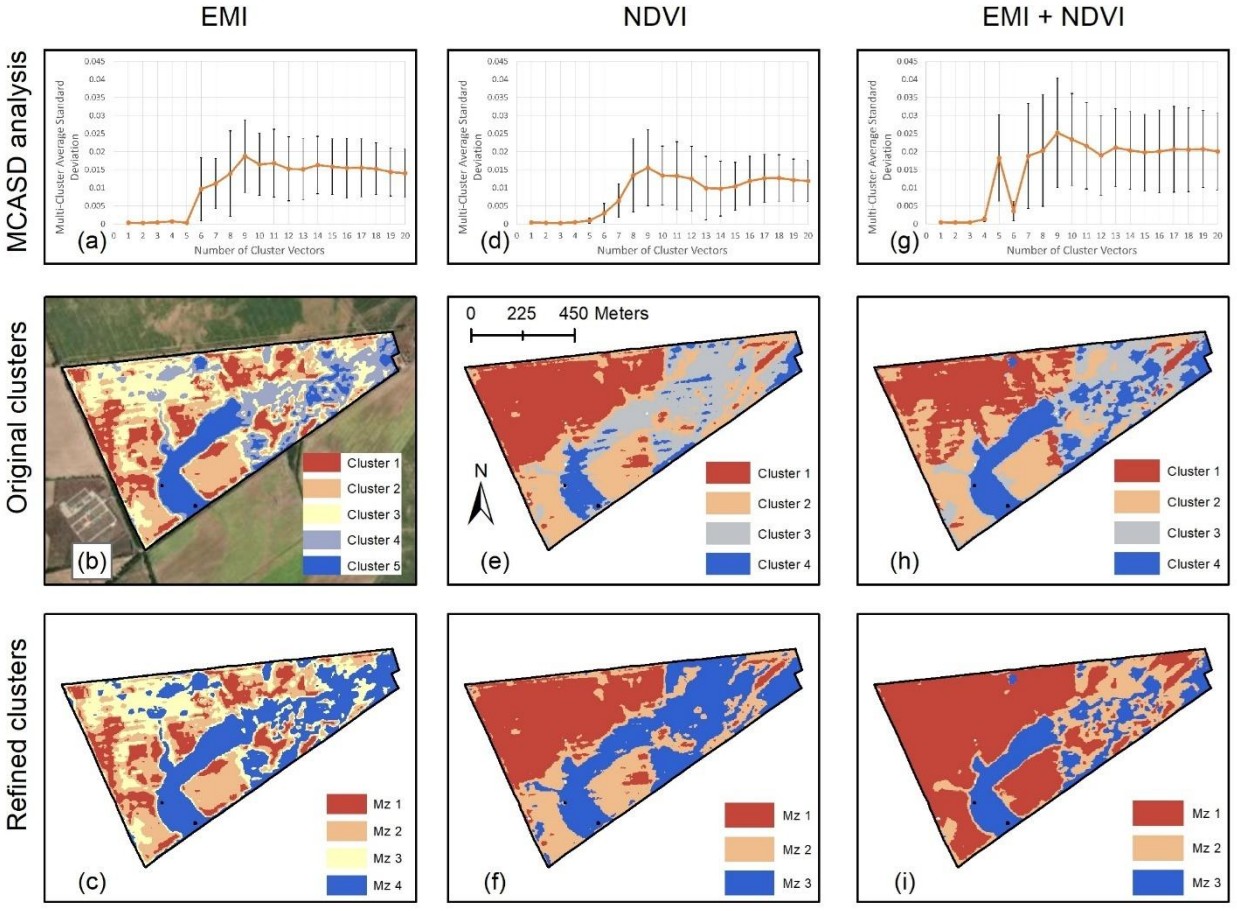


Figure 7. Clustering results for the PatchCROP experimental site. **(a)** MCASD analysis showing
appropriate cluster numbers for EMI data. **(b)** Spatial distribution of original EMI clusters (ESRI,
2020). **(c)** Spatial distribution of refined EMI clusters after post-hoc analysis. **(d)** MCASD analysis
for NDVI data. **(e)** Spatial distribution of original NDVI clusters. **(f)** Spatial distribution of refined
NDVI clusters after post-hoc analysis. **(g)** MCASD analysis for the combined (EMI + NDVI)
dataset. **(h)** Spatial distribution of the original clusters based on the EMI and NDVI data. **(i)** Spatial
distribution of the refined clusters for the combined dataset after post-hoc analysis.



### 3.3 Post-hoc analysis

Starting from the optimal number of clusters identified with MCASD, a post-hoc analysis based on the nine available yield maps and the point-scale soil samples was conducted. The aim was to verify that the cluster are not only statistically separated in terms of the input data (i.e., EMI, NDVI or a combination of EMI and NDVI), but also in terms of yield and soil characteristics (i.e., texture of the first and second layers, depth to the second layer). For the EMI-based clusters, 18 soil sampling locations were within Cluster 4 and only four of these had an EOS layer within 100 cm depth. The other 14 locations had EOS layer below the sampling depth of 100 cm and thus no textural values for the lower layer. Thus, the EOS layer depth of Cluster 4 was assumed to be below 100 cm, and the texture of the lower layer was excluded from further analysis to have a more consistent characterization of the prevailing soil characteristics.

Post-hoc analysis indicated that not all clusters were significantly different from each other, either in terms of yield or soil characteristics. Based on the results of the post-hoc analysis, clusters were either left separated when yield or soil characteristics were statistically different ($p < 0.05$) or grouped together when no statistical separation was identified. For example, Clusters 1, 2, and 3 of the EMI-based classification had at least one significant difference in texture, EOS layer, or yield. On the contrary, cluster 4 and 5 did not show statistically significant differences for any of the investigated properties. Thus, Cluster 4 and 5 were merged together and the resulting EMI-based cluster map had four clusters with statistically significant separation of input data (i.e., EMI), yield, and soil characteristics. A more detailed breakdown of this post-hoc analysis and the resulting merging decisions is provided in Appendix B.

After this post-hoc analysis, the resulting refined maps (Figure 7c, f and i) now have clusters that
are statistically separated in terms of the input dataset (i.e., EMI and NDVI) but also in terms of
the target variables, which are yield and soil characteristics. Therefore, they are referred to as
management zones instead of clusters from this point onwards. These management zone maps
appear to be a simplification of the original clustered maps (Figure 7b, e and h), but they now
provide a more holistic understanding of the field by integrating below-ground (EMI) and above-
ground (NDVI) information with yield and soil data.

**3.4 Assessment of management zones derived from different datasets**
For each management zone of the maps derived from EMI, NDVI, and a combination of EMI-
NDVI, Table 3 shows the average yield between 2011 and 2019 and average soil characteristics,
specifically the depth of the soil texture transition (EOS) and the textural fractions (percentages of
sand, silt, and clay) of two layers up to 100 cm depth. The average yields of Table 3 vary
considerably between different years and follow a general trend of decreasing yields with
increasing cluster number. Thus, yields decrease with decreasing $ECa_z$ and NDVI.

Table 3. Average values of yield (dt/ha) and soil properties for the management zones (MZs)
derived from EMI, NDVI, and a combination of EMI and NDVI.

| | | | EMI | | | | NDVI | | | EMI-NDVI | | |
|---|---|---|---|---|---|---|---|---|---|---|---|---|
| | | **MZs** | **1** | **2** | **3** | **4** | **1** | **2** | **3** | **1** | **2** | **3** |
| **Yield** | | **2011** | 49.5 | 44.7 | 46.5 | 31.7 | 55.9 | 41.1 | 27.5 | 50.7 | 33.1 | 25.7 |
| | | **2012** | 53.4 | 53.1 | 52.6 | 38 | 57.9 | 52.2 | 34.4 | 56.2 | 41.2 | 32.6 |
| | | **2013** | 106.3 | 105.6 | 106.5 | 98.1 | 111.1 | 104.9 | 94.49 | 108.8 | 99.1 | 93.4 |
| | | **2014** | 86.4 | 83.9 | 86.3 | 72.5 | 95.3 | 78.5 | 69.0 | 89.3 | 72.5 | 67.8 |
| | | **2015** | 55.1 | 53.7 | 51.0 | 28.5 | 62.9 | 50.1 | 22.2 | 59.1 | 31.1 | 20.5 |
| | | **2016** | 94.0 | 93.1 | 90.2 | 62.3 | 108.5 | 85.2 | 53.4 | 101 | 61.4 | 53.0 |
| | | **2017** | 78.7 | 76.0 | 73.7 | 47.9 | 89.4 | 69.4 | 41.0 | 83.3 | 48.5 | 39.5 |
| | | **2018** | 40.3 | 39.6 | 38.8 | 26.9 | 44.8 | 37.6 | 23.7 | 42.6 | 29.0 | 21.9 |
| | | **2019** | 71.0 | 69.1 | 67.2 | 48.1 | 80.2 | 62.5 | 43.1 | 74.6 | 47.7 | 42.2 |
| **Soil characteristics** | **Layer 1 (above EOS)** | **Sand %** | 68.2 | 72.4 | 78.1 | 86.2 | 68.6 | 79.5 | 87.2 | 69.8 | 88.4 | 85.2 |
| | | **Silt %** | 23.3 | 20.0 | 16.1 | 9.6 | 23.0 | 15.2 | 8.9 | 22.2 | 8.1 | 10.4 |
| | | **Clay %** | 8.5 | 6.9 | 5.7 | 4.1 | 8.0 | 5.2 | 3.8 | 7.7 | 3.4 | 4.3 |
| | | **Depth (cm)** | 54.0 | 66.9 | 73.1 | 100 | 62.7 | 71.0 | 87.4 | 63.8 | 77.0 | 100 |
| | **Layer 2 (below EOS)** | **Sand %** | 58.3 | 58.0 | 60.6 | NA | 58.1 | 57.8 | 66.1 | 58.1 | 64.9 | NA |
| | | **Silt %** | 23.0 | 23.2 | 21.9 | NA | 23.1 | 23.1 | 19.3 | 23.1 | 19.9 | NA |
| | | **Clay %** | 18.6 | 18.7 | 17.5 | NA | 18.7 | 19.0 | 14.5 | 18.8 | 15.1 | NA |


Figure 8 shows the variation in rye yield (dt/ha) for the management zones derived from different
data sources for the year 2019, which is considered representative for most previous years while
also allowing a direct comparison with the NDVI data for the 2019 growing season. For the EMI-
based management zones (Figure 8a), the yield distributions for the zones 1-3 are relatively similar,
with overlapping interquartile ranges and medians. This indicates that the EMI-based management
zones are more reflective of subsurface soil properties than yield variability for this particular field.
However, zone 4 showed significantly lower yields, corresponding to sandy soils with poor
moisture retention (see Table 3). The NDVI-based management zones (Figure 8b) demonstrate
stronger differentiation in yield distribution and a more consistent decline in yield between zones,
reflecting the ability of NDVI to capture above-ground vegetation vigour and crop health. In
particular, zone 2 reflects an intermediate yield zone between zone 1 and 3, showcasing the ability
of NDVI to differentiate changes in crop performance. The management zones derived from
combining EMI and NDVI (Figure 8c) offer narrower interquartile ranges, particularly in zone 2,
compared to NDVI-based management zones. This indicates that the integration of EMI and NDVI
provides a more consistent and stable representation of yield variability, combining subsurface soil
properties with above-ground dynamics. Although NDVI alone offers slightly more pronounced
yield differentiation, the combined dataset balances both subsurface and vegetation-related factors
effectively, making it a robust approach for management zone delineation. Similar boxplots for
additional years are provided in Appendix C.


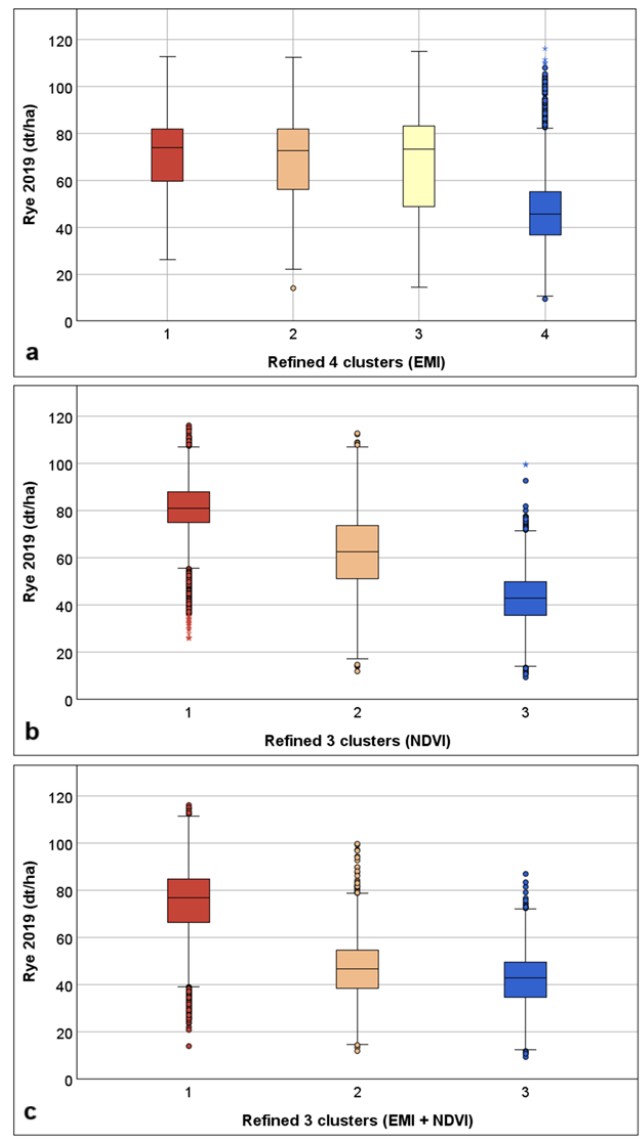

Figure 8. Boxplots illustrating rye yield (dt/ha) for 2019 across management zones (MZs)

derived from **(a)** EMI, **(b)** NDVI, and **(c)** a combination of EMI and NDVI datasets.


The refined management zones can be associated with a typical soil profile based on the average

soil characteristics (Figure 9). The soil profiles show the textural properties of the first two soil

layers and the depth of the interface between these layers (EOS) up to a depth of 100 cm. In some

profiles, the EOS layer reaches 100 cm, and thus the textural properties of the second layer are not

available. In case of the EMI-based zones (Figure 9a-b), zone 1 is characterized by generally higher
$ECa_z$ values, and identifies areas with a substantial average clay content, especially in the second
soil layer (18.6%). Moreover, the sandier top layer is rather shallow and reaches depth of around
54 cm. Moving from zone 1 to zone 4, $ECa_z$ generally decreases. At the same time, the depth of
the top layer (EOS) becomes deeper while the clay and silt content of the soil decreases and the
sand content increases. In zone 4, the average clay content up to 100 cm is 4.1%, while the sand
content is 86.2%. In the case of the NDVI-based management zones (Figure 9c-d), the three zones
appear to be more indicative of crop development, which results in typical soil profiles with
differences that seem less pronounced compared to the case of EMI-based zonation. In this case,
NDVI is generally higher in Cluster 1 and lowest in Cluster 3. The change in soil characteristics
between zones follows a similar trend compared to that of EMI-based zones. The depth of the
interface between soil layer 1 and 2 increases from 62.7 to 87.4 cm from zone 1 to 3, while the
sand content of both layers also increases (from 68.6 to 87.2 % and 58.1 to 66.1 %, respectively).
The management zones derived from the combined EMI-NDVI dataset (Figure 9e-f) have typical
soil profiles that are similar to those based on NDVI. Also, the sand, silt, and clay content of the
first soil layer appear to be rather similar. However, the range of the depth of the interface between
soil layer 1 and 2 is higher for the EMI-NDVI clustered map (63.8 to 100 cm) compared to that of
NDVI-based profiles (62.7 to 87.4 cm). At the same time, the difference in texture between the
second soil layer of Clusters 1 and 2 is stronger in the profiles based on a combination of EMI and
NDVI data (see Table 3). These two factors show that the management zones from EMI and NDVI
have a relatively high variation between soils of different management zones, which is an
improvement compared to the case of the NDVI-based management zones.

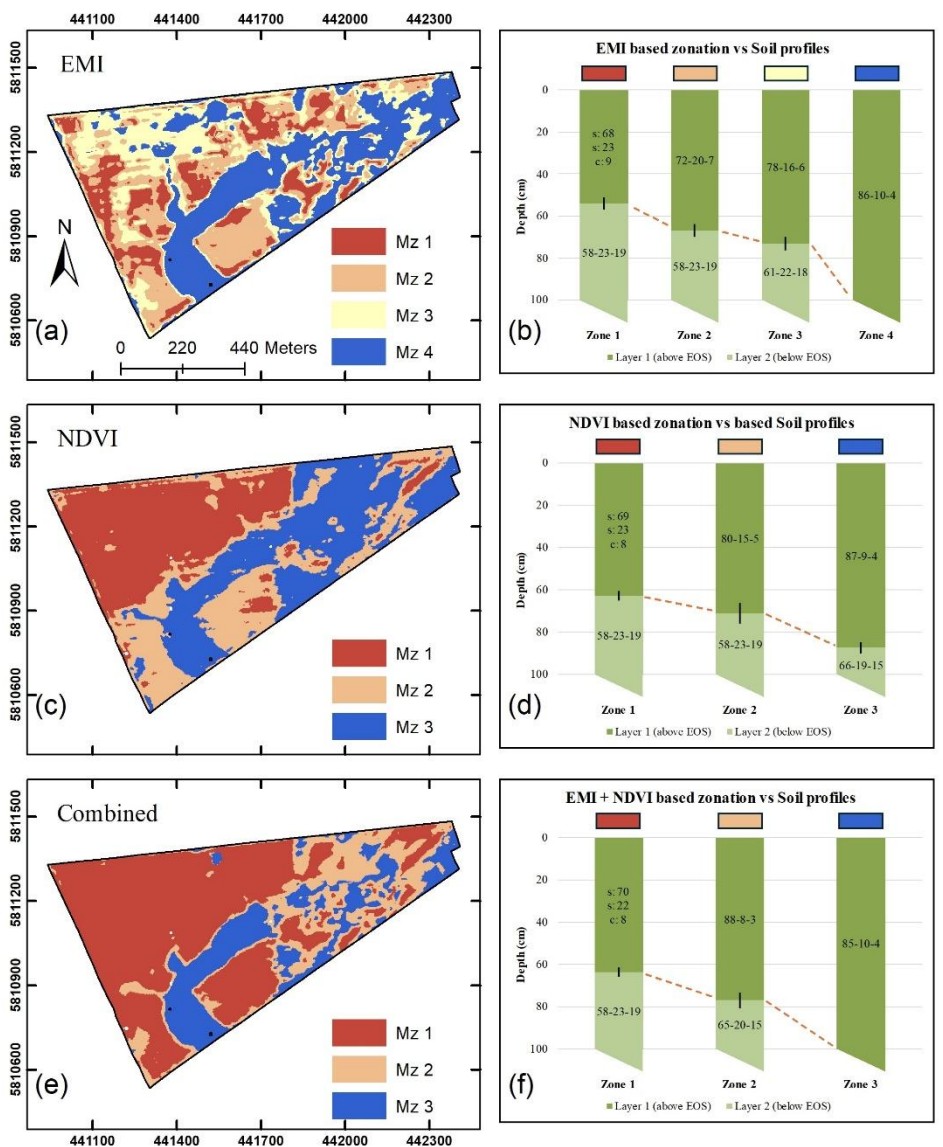

Figure 9. Final management zone maps derived from **(a)** EMI, **(c)** NDVI, and **(e)** a combination of EMI and NDVI datasets. Each zone represents areas with similar subsurface and/or above-ground characteristics. **(b, d, f)** Corresponding soil profiles for each management zone, detailing soil texture (sand-silt-clay %), dotted lines between zones indicate depth of textural change (Layer 1: above EOS; Layer 2: below EOS) and error bar represents the standard error.

In a final step, statistical validation of the management zones was conducted using pairwise t-tests
to evaluate the degree of significant differences in yield and soil properties across consecutive
zones. The results are summarized in Table 4. A pairwise t-test for neighbouring zones derived
from EMI indicated that the yield of 2012, 2013, and 2016 was not significantly different between
zone 1 and zone 2 ($p = 0.603, 0.060, 0.253$) while the yield of 2012 was not significantly different
between zone 2 and 3 ($p = 0.209$). All other pairwise comparisons indicated significant differences
in mean yield. The textural composition of layer 1 was significantly different between all EMI-
derived zones. On the contrary, the depth of top layer was not significantly different between zone
2 and 3 ($p = 0.167$). In addition, the composition of soil layer 2 was not significantly different
between zone 1 and 2 ($p$ of 0.498 for sand, 0.636 for silt, and 0.805 for clay).

The pairwise t-tests between neighbouring zones based on NDVI indicated that differences in yield
among all investigated years were statistically significant. On the contrary, both the depth of the
top layer and the composition of soil layer 2 were not significantly different between zone 1 and 2
($p$ of 0.147 for depth, 0.558 for sand, 0.986 for silt, and 0.627 for clay). These results show that
EMI-based zones subdivided the area in one additional class and provided a more comprehensive
representation of soil properties up to 100 cm compared to the NDVI-based zones for the
investigated field. At the same time, the NDVI-based zones offered a better representation of yield
from 2011 to 2019.

The pairwise t-test between neighbouring zones based on the combined EMI-NDVI dataset
showed that the three zones were significantly different for both yield and soil characteristics. This
indicates that integrating EMI and NDVI datasets allows for the delineation of zones that are robust

651 in representing both yield variability and soil heterogeneity. Moreover, a visual inspection of the

652 management zone maps (Figure 9) shows that both maps based solely on EMI or NDVI are

653 affected by West-East oriented patterns due to measurement direction for EMI and tractor lines in

654 NDVI. These features are not present in the management zone map that integrates EMI and NDVI,

655 suggesting that it also provides a representation of the field that is less affected by external factors.

656 These results underscore the added value of integrating complementary datasets to capture the full

657 spectrum of variability within the field, supporting more informed and effective precision

658 agriculture practices.


 Table 4. Results of the pairwise t-tests for yield and soil properties between management zones

derived from EMI, NDVI, and EMI-NDVI. Bold font indicates significant differences.

|  |  |  | **EMI** |  |  | **NDVI** |  | **EMI - NDVI** |  |
|---|---|---|---|---|---|---|---|---|---|
|  |  | **Cluster** | **1vs2** | **2vs3** | **3vs4** | **1vs2** | **2vs3** | **1vs2** | **2vs3** |
| **Yield** |  | **2011** | **< 0.001** | **< 0.001** | **< 0.001** | **< 0.001** | **< 0.001** | **< 0.001** | **< 0.001** |
|  |  | **2012** | 0.603 | 0.209 | **< 0.001** | **< 0.001** | **< 0.001** | **< 0.001** | **< 0.001** |
|  |  | **2013** | 0.060 | **0.008** | **< 0.001** | **< 0.001** | **< 0.001** | **< 0.001** | **< 0.001** |
|  |  | **2014** | **< 0.001** | **< 0.001** | **< 0.001** | **< 0.001** | **< 0.001** | **< 0.001** | **< 0.001** |
|  |  | **2015** | **0.007** | **< 0.001** | **< 0.001** | **< 0.001** | **< 0.001** | **< 0.001** | **< 0.001** |
|  |  | **2016** | 0.253 | **< 0.001** | **< 0.001** | **< 0.001** | **< 0.001** | **< 0.001** | **< 0.001** |
|  |  | **2017** | **< 0.001** | **0.002** | **< 0.001** | **< 0.001** | **< 0.001** | **< 0.001** | **< 0.001** |
|  |  | **2018** | **0.039** | **0.007** | **< 0.001** | **< 0.001** | **< 0.001** | **< 0.001** | **< 0.001** |
|  |  | **2019** | **0.002** | **0.003** | **< 0.001** | **< 0.001** | **< 0.001** | **< 0.001** | **< 0.001** |
| **Soil** | **Layer 1 (above EOS)** | **Sand %** | **< 0.001** | **0.001** | **< 0.001** | **< 0.001** | **< 0.001** | **< 0.001** | **< 0.001** |
|  |  | **Silt %** | **< 0.001** | **0.006** | **< 0.001** | **< 0.001** | **< 0.001** | **< 0.001** | **< 0.001** |
|  |  | **Clay %** | **< 0.001** | **0.014** | **< 0.001** | **< 0.001** | **< 0.001** | **< 0.001** | **< 0.001** |
|  |  | **Depth (cm)** | **0.004** | 0.167 | **NA** | 0.147 | **0.004** | **0.002** | **NA** |
|  | **Layer 2 (below EOS)** | **Sand %** | 0.498 | **0.010** | NA | 0.558 | **< 0.001** | **< 0.001** | NA |
|  |  | **Silt %** | 0.636 | **0.009** | NA | 0.986 | **0.004** | **0.003** | NA |
|  |  | **Clay %** | 0.805 | 0.056 | NA | 0.627 | **< 0.001** | **< 0.001** | NA |


## 3.5 Limitations and perspectives for future work

This study successfully demonstrated the integration of EMI and NDVI datasets for the delineation
of management zones, but some limitations are still present and should be addressed in future
research. The EMI data were collected during different campaigns under varying environmental
conditions (e.g., soil temperature and moisture) and thus required z-score normalization to
minimize variability. While effective in this study, this approach may not fully account for certain
external factors such as the impact of different management practices in different parts of the field.
Similarly, the NDVI dataset was limited to the 2019 growing season as a) PlanetScope imagery
became accessible for this region only in 2019 and b) the subdivision of the field in differently
cultivated patches from 2020 prevented the use of later satellite products. Nonetheless, the choice
of PlanetScope imagery (3 m resolution) enabled to capture detailed within-field variability in
NDVI, which was particularly important in this study area due to the spatial heterogeneity
introduced by soil variation. If coarser-resolution imagery such as Sentinel-2 (10 m) were used
instead, smaller-scale patterns in crop development or soil-related variation would have been less
detectable due to spatial averaging. This could have reduced the effectiveness of the SOM
clustering in identifying distinct management zones. However, for more homogeneous or large-
scale fields, Sentinel-2 data could be a practical and freely accessible alternative (Kaya et al.,
2025). Another limitation of this study is that the 2019 dataset was considered to be representative
of the investigated area. However, a single season of NDVI data may not fully capture interannual
variability driven by climatic conditions or crop management practices (Scudiero et al., 2018).
Incorporating NDVI data from multiple years in future studies could enable a more comprehensive
analysis of temporal dynamics and their impact on management zone delineation to capture yield
and soil variability.

A further limitation of the study design was the distribution of soil sampling locations. Although
the 160 sampling points provided valuable insights, leveraging EMI-based maps to guide targeted
soil sampling could improve spatial representativeness. Additionally, while EMI in this study had
a depth of investigation of up to 270 cm, soil sampling was limited to 100 cm depth, potentially
missing soil heterogeneity that can affect crops.

Regarding data process, min-max scaling was a suitable method in this study due to the relatively smooth and filtered input data, both for EMI and NDVI. However, this scaling approach is known to be sensitive to outliers and data range extremes (Pedregosa et al., 2011). For datasets with greater variability or different preprocessing methods, alternative scaling approaches such as standardization or robust scaling could be more appropriate (de Amorim et al., 2023). Another factor was the proper application of data normalization prior to clustering, which was essential for obtaining meaningful results in this study (see Appendix A). Future studies should assess the impact of different scaling and normalization strategies on clustering outcomes, especially in settings with noisier or unfiltered sensor data.

702

In this study, clustering relied on a combination of Multi-Cluster Average Standard Deviation (MCASD) to determine the optimal number of clusters and self-organized maps (SOM). While cluster variability was addressed using the Multi-Cluster Average Standard Deviation (MCASD) across 100 SOM runs to a large extent, future studies may benefit from incorporating additional stability metrics such as the Adjusted Rand Index (ARI) or cluster overlap measures to better assess classification consistency. The availability of yield and soil data supported the refinement of SOM-based clusters, enabling the merging of groups that were not agronomically distinct. These datasets helped to ensure that the final management zones were both data-driven and interpretable. However, in scenarios where such ground-truth data are limited or unavailable, the initial clusters may still offer useful insights, albeit with greater uncertainty in their agronomic interpretation. Thus, the presented post-hoc validation step added confidence in the results, but is not strictly required.


The SOM algorithm and the statistical methods used in this study (ANOVA, Tukey's HSD, and t-
tests) do not explicitly account for spatial autocorrelation, which is inherently present in the
interpolated geospatial datasets used here. This may influence statistical outcomes or lead to less
spatially coherent clusters in some cases. For instance, kriging interpolation introduces a spatial
structure that may challenge the assumption of independence underlying post-hoc statistical tests.
However, the use of multi-year yield trends and high-resolution soil data helped reduce uncertainty
in post-hoc validation. Future studies may benefit from incorporating spatially explicit methods,
such as spatially constrained clustering, variogram-based diagnostics, or spatial ANOVA, to better
account for spatial dependence during both classification and validation stages. In addition to these
methodological considerations, future studies should focus on improving the temporal consistency
of data collection and increasing the density and depth of soil sampling. The quantification of
uncertainty in management zone delineation could be also investigated, for example through
ensemble clustering or by incorporating uncertainty from spatial inputs such as EMI interpolation.
Finally, long-term monitoring using datasets from multiple years could provide insights into the
temporal stability of management zones and their relationship with yield.

The detailed management zone maps complemented with soil characterization obtained in this
study should in a next step be integrated into agroecosystem models. This well enable to simulate
and predict the impact of different management strategies under future environmental and climatic
conditions, and thus help to optimize irrigation, fertilization, and other field management practices,
further supporting decision-making for sustainable and resource-efficient agriculture.

## 4 Conclusions

This study integrated proximal soil sensing (EMI) and remote sensing (NDVI) data to delineate high-resolution management zones in a 70 ha agricultural field. Self-Organizing Maps (SOM), an advanced unsupervised machine learning technique, were combined with statistical validation methods to identify spatial areas with similar above- and below-ground properties. Historical yield maps and detailed soil information up to a depth of 100 cm were used to refine and validate the clustering results, ensuring both their accuracy and practical applicability.

To address the variability introduced by environmental conditions during data collection, EMI measurements from multiple campaigns were standardized using z-score normalization, ensuring consistent input for further analysis of the investigated field. Similarly, NDVI data from the 2019 growing season were selected as they represented an uninterrupted crop cycle prior to the subdivision of the investigated field in multiple patches. Before clustering, data was appropriately normalized. The Multi-Cluster Average Standard Deviation (MCASD) method was applied to determine the optimal number of clusters for different datasets. The optimal number of clusters was determined to be five using the EMI data, four for the NDVI data, and four for the combination of EMI and NDVI datasets. However, statistical validation through Tukey's post-hoc analysis using independent yield maps and soil samples reduced the clusters number to 4, 3, and 3, respectively. This ensured that the clusters were not only computationally distinct with respect to the input data, but also significantly different in terms of soil characteristics and yield data, thereby increasing their practical relevance in precision agriculture.

Results showed that EMI-based management zones provided a better representation of subsurface
properties, particularly soil texture and the depth at which textural changes occur, which underlines
the utility of EMI for guiding soil management practices. In comparison, NDVI-based
management zones aligned more closely with topsoil characteristics and yield maps, effectively
capturing above-ground variability. In general, the integration of EMI and NDVI datasets provided
a more comprehensive representation of the spatial variability of both soil characteristics and yield,
resulting in management zones that linked both subsurface soil conditions and above-ground
vegetation performance. These combined zones effectively explained productivity patterns by
bridging the gap between soil properties and crop health.

The product of this study is a high-resolution management zonation map which would provide a
significant added value in precision and sustainable agriculture. Moreover, it can help in setting-
up of agroecosystem models for the simulation of crop performance and yield and in guiding future
soil sampling campaigns. Finally, the workflow proposed in this study can provide a robust
blueprint for unsupervised clustering of proximal soils sensing and remote sensing data in
agriculture, and future studies should explore the scalability of this methodology in different
climatic conditions or other crop systems, as well as investigate additional data sources to further
enhance the representation of within-field heterogeneity in soil and crops.

**Appendix A: Influence of data normalization**

Figure A1 shows a visual comparison of management zone delineation using different normalization approaches. These are: a) EMI-based clustering of ECa$_z$ maps, b) combined EMI-NDVI clustering with dataset-wise normalization (i.e., normalized by using the minimum and maximum values for all the available data), and c) combined EMI-NDVI clustering with dataset-wise normalization of EMI data and separate column-wise normalization of NDVI data. As apparent in Figure A1b, the EMI measurements dominate the clustering results when an inappropriate normalization is used. On the contrary, the normalization strategy used here (Figure A1c) provides a clustering result where both EMI and NDVI meaningfully contribute.

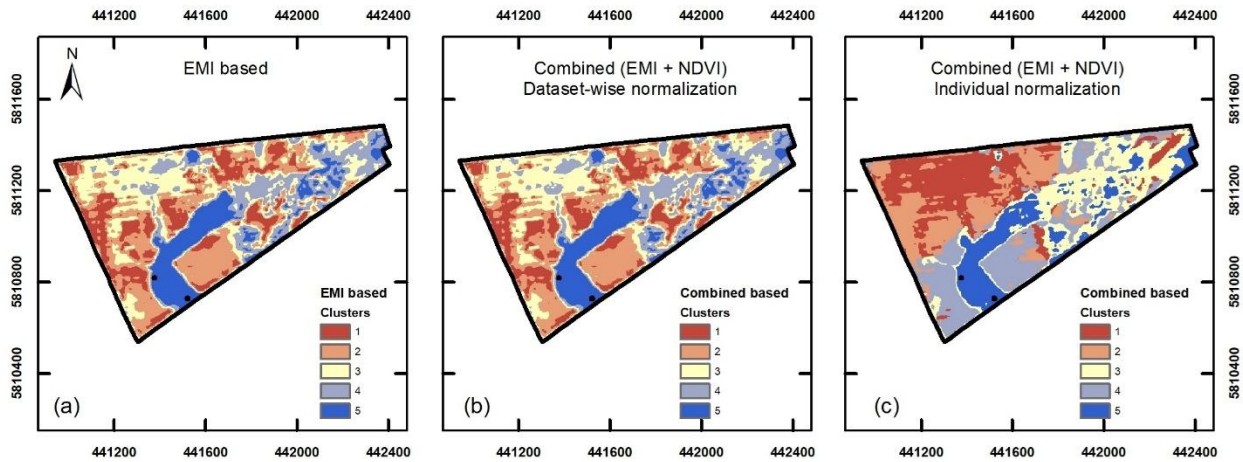

Figure A1. Comparison of management zone delineation using different normalization approaches **(a)** EMI-based clustering without normalization, **(b)** Combined EMI and NDVI clustering with dataset-wise normalization, **(c)** Combined EMI and NDVI clustering with individual normalization, where EMI data were normalized as a dataset, while NDVI data were normalized column-wise.

**Appendix B: Additional results for post-hoc analysis**

For the EMI dataset (VCP + HCP, 9 coils), the MCASD analysis suggested five clusters. The results of the post-hoc analysis are shown in Table B1. Statistically significant differences between two clusters are indicated by an *O* whereas an *X* indicates no significant differences. When two clusters have no statistically significant difference for any of the evaluated properties, they are merged. Therefore, clusters 4 and 5 were merged into a new cluster 4. For the NDVI dataset, the MCASD analysis suggested 4 clusters and the results of the post-hoc analysis (Table B2) merged clusters 3 and 4 into a new cluster 3. For the combined dataset (EMI + NDVI), the MCASD analysis suggested 4 clusters and the results of the post-hoc analysis (Table B3) merged clusters 1 and 2 into a new cluster 1.

Table B1. Post-hoc analysis of soil characteristics and yield for the EMI-based clusters leading to cluster merging. Statistically significant *(O)* or non-significant differences *(X)* are provided between clusters for soil texture, EOS layer, and yield.

| Clusters | | 1vs2 | 2vs3 | 3vs4 | 4vs5 |
|---|---|---|---|---|---|
| **End of sandy layer (Depth cm)** | | *O* | *X* | *O* | *X* |
| **Layer 1 (above EOS)** | **Sand** | *X* | *O* | *O* | *X* |
| | **Silt** | *X* | *O* | *O* | *X* |
| | **Clay** | *X* | *O* | *O* | *X* |
| **Layer 2 (below EOS)** | **Sand** | *X* | *X* | *O* | *X* |
| | **Silt** | *X* | *X* | *O* | *X* |
| | **Clay** | *X* | *X* | *O* | *X* |
| **Yield** | | *X* | *X* | *O* | *X* |

Table B2. Post-hoc analysis of soil characteristics and yield for the NDVI-based clusters leading
to cluster merging. Statistically significant *(O)* or non-significant differences *(X)* are provided
between clusters for soil texture, EOS layer, and yield.

| Clusters | | 1vs2 | 2vs3 | 3vs4 |
|---|---|---|---|---|
| **End of sandy layer (depth cm)** | | *X* | *O* | *X* |
| **Layer 1 (above EOS)** | **Sand** | *O* | *O* | *X* |
| | **Silt** | *O* | *O* | *X* |
| | **Clay** | *O* | *O* | *X* |
| **Layer 2 (below EOS)** | **Sand** | *X* | *O* | *X* |
| | **Silt** | *X* | *O* | *X* |
| | **Clay** | *X* | *O* | *X* |
| **Yield** | | *X* | *O* | *X* |


Table B3. Post-hoc analysis of soil characteristics and yield for the clusters based on EMI and
NDVI leading to cluster merging. Statistically significant *(O)* or non-significant differences *(X)*
are provided between clusters for soil texture, EOS layer, and yield.

| Clusters | | 1vs2 | 2vs3 | 3vs4 |
|---|---|---|---|---|
| **End of sandy layer (depth cm)** | | *X* | *O* | *O* |
| **Layer 1 (above EOS)** | **Sand** | *X* | *O* | *O* |
| | **Silt** | *X* | *O* | *O* |
| | **Clay** | *X* | *O* | *O* |
| **Layer 2 (below EOS)** | **Sand** | *X* | *O* | *X* |
| | **Silt** | *X* | *O* | *X* |
| | **Clay** | *X* | *O* | *X* |
| **Yield** | | *X* | *O* | *X* |


**Appendix C: Differences in yield between derived management zones**

Figure C1 presents boxplots illustrating yield variability (dt/ha) for two additional years: winter rye in 2017 (Figure C1a) and rapeseed in 2018 (Figure C1b). Results are presented for management zones derived from three clustering approaches: EMI-based (left), NDVI-based (middle), and combined EMI + NDVI (right). These two years were selected as additional representative examples, as the overall yield variation across the full nine-year dataset followed the same trend. In the EMI-based management zones, yield distribution is relatively similar across the first three zones, with a noticeable drop in the fourth zone. In contrast, NDVI-based and EMI + NDVI zones show a progressive decline in yield across clusters, indicating a clearer trend of decreasing productivity.

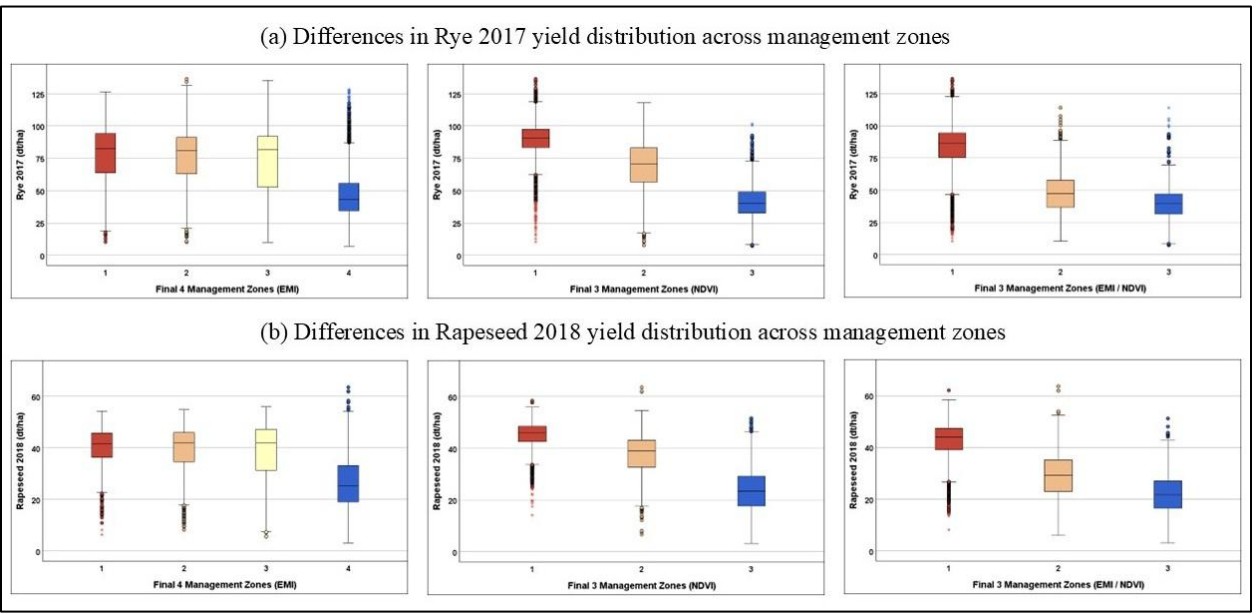

Figure C1. Yield distribution across final management zones based on EMI, NDVI, and combined EMI-NDVI datasets.

**Data availability**

The data that support the findings of this study are available on request from the corresponding author.

**Author contributions**

SD, CB, and JH: conceptualization and methodology; SD, CB, MD, and IO: field measurements; SD, MD, DL and CB: data analysis; SD: writing – original draft; CB, DL, IO, MD, HV, and JH: writing: review and editing; JH – project supervision. All authors have read and agreed to the published version of the manuscript.

**Competing interest**

A co-author (Dave O'Leary) of this article is a member of the guest editorial board for this Special Issue: "Agrogeophysics: illuminating soil's hidden dimensions".

**Special issue statement**

This article is part of the special issue "Agrogeophysics: illuminating soil's hidden dimensions". It is not associated with a conference.

**Acknowledgements**

We thank Dr. agr. Kathrin Grahmann, Robert Zieciak, Anna Engels, Tawhid Hossain for local support, organizational formalities and data. Danial Mansourian, Ali Chaudhry, Emilio Capitanio, Dr. Muhammad Fahad, and Ali Sadrzadeh are thanked for their help during the EMI measurement

campaigns. The maintenance of the patchCROP infrastructure is supported by the Leibniz Centre
for Agricultural Landscape Research.

**Financial support**
This research was supported by the DFG (German Research Foundation) through the Germany's
Excellence Strategy EXC 2070, Grant No. 390732324 - PhenoRob.

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

Index Based on High Spatial Resolution Satellite Images Reveals Insight-Driven Edaphic
Management Zones, AgriEngineering, 7, https://doi.org/10.3390/agriengineering7040092, 2025.

Keller, G. . and Frischknecht, F. .: Electrical Methods in Geophysical Propecting, Oxford, New
York, Pergamon Press, 1966.

Khan, S., Tufail, M., Khan, M. T., Khan, Z. A., Iqbal, J., and Alam, M.: A novel semi-supervised
framework for UAV based crop/weed classification, PLoS One, 16,
https://doi.org/10.1371/journal.pone.0251008, 2021.

Kibblewhite, M. G., Ritz, K., and Swift, M. J.: Soil health in agricultural systems, Philos. Trans.
R. Soc. B Biol. Sci., 363, 685–701, https://doi.org/10.1098/rstb.2007.2178, 2008.

Koch, T., Deumlich, D., Chifflard, P., Panten, K., and Grahmann, K.: Using model simulation to
evaluate soil loss potential in diversified agricultural landscapes, Eur. J. Soil Sci., 74, 1–14,
https://doi.org/10.1111/ejss.13332, 2023.

Koganti, T., De Smedt, P., Farzamian, M., Knadel, M., Triantafilis, J., Christiansen, A. V., and
Greve, M. H.: Editorial: Digital soil mapping using electromagnetic sensors, Front. Soil Sci., 4,
10–12, https://doi.org/10.3389/fsoil.2024.1536797, 2024.

Kohonen, T.: Essentials of the self-organizing map, Neural Networks, 37, 52–65,
https://doi.org/10.1016/j.neunet.2012.09.018, 2013.

Kuang, B., Mahmood, H. S., Quraishi, M. Z., Hoogmoed, W. B., Mouazen, A. M., and van

Henten, E. J.: Sensing Soil Properties in the Laboratory, In Situ, and On-Line, 155–223,
https://doi.org/10.1016/B978-0-12-394275-3.00003-1, 2012.
Lavoué, F., Van Der Kruk, J., Rings, J., André, F., Moghadas, D., Huisman, J. A., Lambot, S.,
LWeihermüller, Vanderborght, J., and Vereecken, H.: Electromagnetic induction calibration
using apparent electrical conductivity modelling based on electrical resistivity tomography, Near
Surf. Geophys., 8, 553–561, https://doi.org/10.3997/1873-0604.2010037, 2010.
Li, Y., Ni, Z., Jin, F., Li, J., and Li, F.: Research on Clustering Method of Improved Glowworm
Algorithm Based on Good-Point Set, Math. Probl. Eng., 2018,
https://doi.org/10.1155/2018/8724084, 2018.
Liaghat, S. and Balasundram, S. K.: A review: The role of remote sensing in precision
agriculture, Am. J. Agric. Biol. Sci., 5, 50–55, https://doi.org/10.3844/ajabssp.2010.50.55, 2010.
Liakos, K. G., Busato, P., Moshou, D., Pearson, S., and Bochtis, D.: Machine learning in
agriculture: A review, Sensors (Switzerland), 18, 1–29, https://doi.org/10.3390/s18082674, 2018.
Liang, J., Zhao, X., Li, D., Cao, F., and Dang, C.: Determining the number of clusters using
information entropy for mixed data, Pattern Recognit., 45, 2251–2265,
https://doi.org/10.1016/j.patcog.2011.12.017, 2012.
Licker, R., Johnston, M., Foley, J. A., Barford, C., Kucharik, C. J., Monfreda, C., and
Ramankutty, N.: Mind the gap: How do climate and agricultural management explain the "yield
gap" of croplands around the world?, Glob. Ecol. Biogeogr., 19, 769–782,
https://doi.org/10.1111/j.1466-8238.2010.00563.x, 2010.
López-Granados, F.: Weed detection for site-specific weed management: mapping and real-time
approaches, Weed Res., 51, 1–11, https://doi.org/10.1111/j.1365-3180.2010.00829.x, 2011.
Lueck, E. and Ruehlmann, J.: Resistivity mapping with GEOPHILUS ELECTRICUS -

Information about lateral and vertical soil heterogeneity, Geoderma, 199, 2–11,

https://doi.org/10.1016/j.geoderma.2012.11.009, 2013.

McNeill, J. D.: Electromagnetic Terrain Conductivity Measurement at Low Induction Numbers,

https://geonics.com/pdfs/technicalnotes/tn6.pdf, 1980.

Meyer, S., Kling, C., Vogel, S., Schröter, I., Nagel, A., Kramer, E., Gebbers, R., Philipp, G.,

Lück, K., Gerlach, F., Scheibe, D., and Ruehlmann, J.: Creating soil texture maps for precision

liming using electrical resistivity and gamma ray mapping, in: Precision agriculture '19, 539–

546, https://doi.org/10.3920/978-90-8686-888-9_67, 2019.

Mohammed, I., Marshall, M., de Bie, K., Estes, L., and Nelson, A.: A blended census and

multiscale remote sensing approach to probabilistic cropland mapping in complex landscapes,

ISPRS J. Photogramm. Remote Sens., 161, 233–245,

https://doi.org/10.1016/j.isprsjprs.2020.01.024, 2020.

Moshou, D., Bravo, C., Wahlen, S., West, J., McCartney, A., De Baerdemaeker, J., and Ramon,

H.: Simultaneous identification of plant stresses and diseases in arable crops using proximal

optical sensing and self-organising maps, Precis. Agric., 7, 149–164,

https://doi.org/10.1007/s11119-006-9002-0, 2006.

Geologischer Dienst NRW: https://www.gd.nrw.de/.

O'Leary, D., Brown, C., Healy, M. G., Regan, S., and Daly, E.: Observations of intra-peatland

variability using multiple spatially coincident remotely sensed data sources and machine

learning, Geoderma, 430, 116348, https://doi.org/10.1016/j.geoderma.2023.116348, 2023.

O'Leary, D., Brogi, C., Brown, C., Tuohy, P., and Daly, E.: Linking electromagnetic induction

data to soil properties at field scale aided by neural network clustering, Front. Soil Sci., 4, 1–13,

https://doi.org/10.3389/fsoil.2024.1346028, 2024.

Öttl, L. K., Wilken, F., Auerswald, K., Sommer, M., Wehrhan, M., and Fiener, P.: Tillage

erosion as an important driver of in-field biomass patterns in an intensively used hummocky

landscape, L. Degrad. Dev., 32, 3077–3091, https://doi.org/10.1002/ldr.3968, 2021.

Patro, S. G. K. and sahu, K. K.: Normalization: A Preprocessing Stage, Iarjset, 20–22,

https://doi.org/10.17148/iarjset.2015.2305, 2015.

Pedregosa, F., Varoquaux, G., Gramfort, A., Michel, V., Thirion, B., Grisel, O., Blondel, M.,

Müller, A., Nothman, J., Louppe, G., Prettenhofer, P., Weiss, R., Dubourg, V., Vanderplas, J.,

Passos, A., Cournapeau, D., Brucher, M., Perrot, M., and Duchesnay, É.: Scikit-learn: Machine

Learning in Python, J. Mach. Learn. Res., 12, 2825–2830,

https://doi.org/10.48550/arXiv.1201.0490, 2011.

Pedrera-Parrilla, A., Brevik, E. C., Giráldez, J. V., and Vanderlinden, K.: Temporal stability of

electrical conductivity in a sandy soil, Int. Agrophysics, 30, 349–357,

https://doi.org/10.1515/intag-2016-0005, 2016.

Pradipta, A., Soupios, P., Kourgialas, N., Doula, M., Dokou, Z., Makkawi, M., Alfarhan, M.,

Tawabini, B., and Kirmizakis, P.: Precision Agriculture — Part 1 : Soil Applications, Water, 14,

1158., 2022.

Robinet, J., von Hebel, C., Govers, G., van der Kruk, J., Minella, J. P. G., Schlesner, A.,

Ameijeiras-Mariño, Y., and Vanderborght, J.: Spatial variability of soil water content and soil

electrical conductivity across scales derived from Electromagnetic Induction and Time Domain

Reflectometry, Geoderma, 314, 160–174, https://doi.org/10.1016/j.geoderma.2017.10.045, 2018.

Romero-Ruiz, A., O'Leary, D., Daly, E., Tuohy, P., Milne, A., Coleman, K., and Whitmore, A.

P.: An agrogeophysical modelling framework for the detection of soil compaction spatial

variability due to grazing using field-scale electromagnetic induction data, Soil Use Manag., 40,

https://doi.org/10.1111/sum.13039, 2024.
Rudolph, S., van der Kruk, J., von Hebel, C., Ali, M., Herbst, M., Montzka, C., Pätzold, S.,
Robinson, D. A., Vereecken, H., and Weihermüller, L.: Linking satellite derived LAI patterns
with subsoil heterogeneity using large-scale ground-based electromagnetic induction
measurements, Geoderma, 241–242, 262–271, https://doi.org/10.1016/j.geoderma.2014.11.015,

1118    2015.

Saifuzzaman, M., Adamchuk, V., Buelvas, R., Biswas, A., Prasher, S., Rabe, N., Aspinall, D.,
and Ji, W.: Clustering tools for integration of satellite remote sensing imagery and proximal soil
sensing data, Remote Sens., 11, 1–17, https://doi.org/10.3390/rs11091036, 2019.
Saputra, Danny Matthew, SAPUTRA, D., and OSWARI, L. D.: Effect of Distance Metrics in
Determining K-Value in K-Means Clustering Using Elbow and Silhouette Method, 172, 341–
346, https://doi.org/10.2991/aisr.k.200424.051, 2020.
Schmäck, J., Weihermüller, L., Klotzsche, A., von Hebel, C., Pätzold, S., Welp, G., and
Vereecken, H.: Large-scale detection and quantification of harmful soil compaction in a post-
mining landscape using multi-configuration electromagnetic induction, Soil Use Manag., 38,
212–228, https://doi.org/10.1111/sum.12763, 2022.
Schubert, E.: Stop using the elbow criterion for k-means and how to choose the number of
clusters instead, ACM SIGKDD Explor. Newsl., 25, 36–42,
https://doi.org/10.1145/3606274.3606278, 2023.
Scudiero, E., Teatini, P., Manoli, G., Braga, F., Skaggs, T. H., and Morari, F.: Workflow to
establish time-specific zones in precision agriculture by spatiotemporal integration of plant and
soil sensing data, Agronomy, 8, 1–21, https://doi.org/10.3390/agronomy8110253, 2018.
Simpson, D., Lehouck, A., Verdonck, L., Vermeersch, H., Van Meirvenne, M., Bourgeois, J.,
Thoen, E., and Docter, R.: Comparison between electromagnetic induction and fluxgate
gradiometer measurements on the buried remains of a 17th century castle, J. Appl. Geophys., 68,
294–300, https://doi.org/10.1016/j.jappgeo.2009.03.006, 2009.
Sishodia, R. P., Ray, R. L., and Singh, S. K.: Applications of remote sensing in precision
agriculture: A review, Remote Sens., 12, 1–31, https://doi.org/10.3390/rs12193136, 2020.
Skakun, S., Kalecinski, N. I., Brown, M. G. L., Johnson, D. M., Vermote, E. F., Roger, J. C., and
Franch, B.: Assessing within-field corn and soybean yield variability from worldview-3, planet,
sentinel-2, and landsat 8 satellite imagery, Remote Sens., 13, 1–18,
https://doi.org/10.3390/rs13050872, 2021.
Stamford, J. D., Vialet-Chabrand, S., Cameron, I., and Lawson, T.: Development of an accurate
low cost NDVI imaging system for assessing plant health, Plant Methods, 19, 1–19,
https://doi.org/10.1186/s13007-023-00981-8, 2023.
Tagarakis, A., Liakos, V., Fountas, S., Koundouras, S., and Gemtos, T. A.: Management zones
delineation using fuzzy clustering techniques in grapevines, Precis. Agric., 14, 18–39,
https://doi.org/10.1007/s11119-012-9275-4, 2013.
Taşdemir, K., Milenov, P., and Tapsall, B.: A hybrid method combining SOM-based clustering
and object-based analysis for identifying land in good agricultural condition, Comput. Electron.
Agric., 83, 92–101, https://doi.org/10.1016/j.compag.2012.01.017, 2012.
Trivedi, M. B., Marshall, M., Estes, L., de Bie, C. A. J. M., Chang, L., and Nelson, A.: Cropland
Mapping in Tropical Smallholder Systems with Seasonally Stratified Sentinel-1 and Sentinel-2
Spectral and Textural Features, Remote Sens., 15, https://doi.org/10.3390/rs15123014, 2023.
Do you know all 17 SDGs? (2021): https://sdgs.un.org/goals.
Usama, M., Qadir, J., Raza, A., Arif, H., Yau, K. L. A., Elkhatib, Y., Hussain, A., and Al-

Fuqaha, A.: Unsupervised Machine Learning for Networking: Techniques, Applications and

Research Challenges, IEEE Access, 7, 65579–65615,

https://doi.org/10.1109/ACCESS.2019.2916648, 2019.

Valentine, A. and Kalnins, L.: An introduction to learning algorithms and potential applications

in geomorphometry and Earth surface dynamics, Earth Surf. Dyn., 4, 445–460,

https://doi.org/10.5194/esurf-4-445-2016, 2016.

Vogel, S., Gebbers, R., Oertel, M., and Kramer, E.: Evaluating soil-borne causes of biomass

variability in Grassland by remote and proximal sensing, Sensors (Switzerland), 19, 1–16,

https://doi.org/10.3390/s19204593, 2019.

Wang, F., Yang, S., Wei, Y., Shi, Q., and Ding, J.: Characterizing soil salinity at multiple depth

using electromagnetic induction and remote sensing data with random forests: A case study in

Tarim River Basin of southern Xinjiang, China, Sci. Total Environ., 754, 142030,

https://doi.org/10.1016/j.scitotenv.2020.142030, 2021.

Wang, L., Duan, Y., Zhang, L., Rehman, T. U., Ma, D., and Jin, J.: Precise estimation of NDVI

with a simple NIR sensitive RGB camera and machine learning methods for corn plants, Sensors

(Switzerland), 20, 1–15, https://doi.org/10.3390/s20113208, 2020.

Ward, S. H. and Hohmann, G. W.: 4. Electromagnetic Theory for Geophysical Applications, in:

Electromagnetic Methods in Applied Geophysics, Society of Exploration Geophysicists, 130–

311, https://doi.org/10.1190/1.9781560802631.ch4, 1988.

Weiss, M., Jacob, F., and Duveiller, G.: Remote sensing for agricultural applications: A meta-

review, Remote Sens. Environ., 236, 111402, https://doi.org/10.1016/j.rse.2019.111402, 2020.

Wilhelm, W. W., Ruwe, K., and Schlemmer, M. R.: Comparison of three leaf area index meters

in a corn canopy, Crop Sci., 40, 1179–1183, https://doi.org/10.2135/cropsci2000.4041179x,

1182    2000.

Xue, J. and Su, B.: Significant remote sensing vegetation indices: A review of developments and
applications, J. Sensors, 2017, https://doi.org/10.1155/2017/1353691, 2017.
Ylagan, S., Brye, K. R., Ashworth, A. J., Owens, P. R., Smith, H., and Poncet, A. M.: Using
Apparent Electrical Conductivity to Delineate Field Variation in an Agroforestry System in the
Ozark Highlands, Remote Sens., 14, 1–25, https://doi.org/10.3390/rs14225777, 2022.
Zhang, Y. and Wang, Y.: Machine learning applications for multi-source data of edible crops: A
review of current trends and future prospects, Food Chem. X, 19, 100860,
https://doi.org/10.1016/j.fochx.2023.100860, 2023.
Zhu, Q., Lin, H., and Doolittle, J.: Repeated Electromagnetic Induction Surveys for Determining
Subsurface Hydrologic Dynamics in an Agricultural Landscape, Soil Sci. Soc. Am. J., 74, 1750–
1762, https://doi.org/10.2136/sssaj2010.0055, 2010.