# Peer review of "Combining Electromagnetic Induction and Satellite-based NDVI Data for Improved Determination of Management Zones for Sustainable Crop Production Authors Salar Saeed Dogar1\*, Cosimo Brogi1, Dave O'Leary2,3, Ixchel M. Hernández-Ochoa4, Marco Donat5"

_EGUsphere, 2025_

## Referee Comment (RC2)

**Review of: 'Combining Electromagnetic Induction and Remote Sensing Data for Improved Determination of Management Zones for Sustainable Crop Production'**

This paper proposes a proximal and remote sensing data harmonisation framework for input into a Self-organizing map (SOM)-based classification for determining field management zones. It is worthy of publication once the following points are considered and addressed:

1. Materials/Methods: The four sub-sections of section 2.2 need re-ordering to demonstrate the workflow: (1) EMI/EC data, (2) RS/NDVI data, (3) Yield data, (4) Soils data. As only the first two are inputs for the SOM/MCASD clustering. The second two are used to 'validate' and refine the clusters.

2. Materials/Methods: A table would be useful to summarise each of these four datasets and their use in the study. The table can list: (a) the period of collection (e.g., 2011-19 for yield data); (b) whether the patchCROP experiment was in operation or not, (c) data processing steps taken (e.g. kriging or some other interpolation, normalisation etc. – see also that stated in section 3), and (d) whether used for SOM/MCASD inputs or used for the (ANOVA-based) validation of SOM clusters (with subsequent merging of clusters) etc.

3. Results: Maps and workflow narratives should be in this order: (1) EMI/EC data (Figs. 3, 4), (2) RS/NDVI data (Fig. 5), (3) Yield data (Fig. 2), (4) Soils data graphic (new), (5) SOM/MCASD clustering maps of EMI/RS plus refinements via yield/soils (Fig. 6).

4. Limitations: When describing the caveats to the methodology (section 3.5), refer to the new Table suggested in (2) for challenges due to different data collection timeframes, patchCROP, data processing, etc.

5. Limitations: What would be the likely consequences of using free, 10m resolution imagery from sentinel 2 say, to that used with the 3m resolution of Planetscope for the NDVI data?

6. Limitations: More on the sensitivity of the SOM-based clusters and their refinements using yield and soil information – from no data available to that available here (as shown in rows 3 and 4 in Fig.6).

7. Limitations: For the clustering methods described (in the introduction) and the SOM method applied (p.6 to p.7) – none implicitly capture spatial effects, such as spatial autocorrelation. Further, the statistical analyses using ANOVAs/Tukey's HSD and t-tests are similarly non-spatial. What are the consequences of this? What methods could be applied for future work to investigate this?

8. Limitations: Given all the above - something on the capture of uncertainty in the demarcation of the management zones for current and future work?

9. Conclusion: More should be said on the choice made for the proximal sensing and the choice made for the satellite remote sensing. For the former, EMI/EC essentially does soil physics / structure / water, while for latter, NDVI does crop health. This is OK but what of the alternatives? For example, using indices from radar-based missions (e.g., sentinel 1) rather than imagery-based missions (e.g., sentinel 2). Insights on how the choice of sensors will ultimately affect the SOM/MCASD clustering and resultant management zones would be useful. For example, in some cases, the precision management of soil water may be more of a focus than the precision management soil nutrients – each requiring specific sensing technologies, etc. Essentially expand discussions in the introduction (p.5-6) and conclusions.

10. Consider changing the title to either: 'Combining Proximal and Satellite Remote Sensing Data for Improved Determination of Management Zones for Sustainable Crop Production' or 'Combining Electromagnetic Induction and Satellite Sensed NDVI Data for Improved Determination of Management Zones for Sustainable Crop Production' – the former is general, while the latter is specific.

---

## Author Comment (AC1)

**Response to reviewers**

**Response to comments and suggestions from Reviewer 1**

**General comments**

The paper presents a relevant contribution to precision agriculture by coupling NDVI and EMI for management zones' delineation. The methodology is generally sound, especially the use of the SOM and MCASD for cluster optimisation. The study presents a robust workflow that could inform both research and practice. However, some aspects need clarification to improve generalisability and interpretability.

We thank the reviewer for the positive assessment, and are happy to provide the requested clarifications.

**Specific Comments**

- Lines 64-122: The review of EMI and NDVI is largely descriptive. It would be stronger if the authors synthesised how of the previous studies succeed or failed in integrating these data types. I suggest adding a short synthesis paragraph summarising what's missing in prior work and how this study fills the gap.

We thank the reviewer for this helpful suggestion. In response, we have now included the following statement in section 1 (lines 118-128):

"In summary, while previous studies have made important contributions towards integrating EMI and NDVI data for management zone delineation (Corwin and Scudiero, 2019; Ciampalini et al., 2015), the results have been highly dependent on sensor resolution, data timing, and local soil-plant interactions. Some studies demonstrated that EMI alone offers strong insights into soil structure and moisture patterns, and suggested that crop-level responses captured by NDVI can be inconsistent due to seasonal and environmental variability. Others highlighted the value of combining datasets but faced limitations in spatial resolution, ground-truth validation, or field-specific conditions that restricted the precision of zone delineation. This study builds on these efforts by combining high-resolution EMI and NDVI data within a harmonized framework, applying consistent normalization, and validating the resulting zones with multi-year yield data and dense soil sampling."

- Lines 304- 309: the use of min-max scaling prior to clustering is appropriate for ensuring feature comparability. However, the authors should briefly justify this choice over alternatives (e.g., standardisation, robust scaling), especially given the potential presence of outliers in EMI and NDVI data. Min-max is sensitive to extreme values, which may distort the input space and affect cluster geometry in SOM.

We thank the reviewer for pointing out this important consideration. In our study, EMI data were already filtered to remove outliers from a variety of sources. In particular, the combination of min-max filtering, histogram filtering, and ECa variation filtering effectively remove outliers from the EMI data, as shown in previous research. We therefore are confident that the resulting distribution of ECa values, combined with the use of z-transform normalization, is appropriate for a min-max scaling. Similarly, NDVI maps were pre-processed by PlanetScope to remove atmospheric artifacts, and we manually excluded data from periods with low vegetation signal (lines 452-457). Moreover, the extent of the area and the amount of pixels in the NDVI images assures that the distribution is free from outliers that would affect the min-max scaling.

Nonetheless, we understand that this may not be the case in other areas or when different data sources are used. Thus, we now addressed these topics in section 3.5, where the new text reads:

"Although min-max scaling was suitable in this study due to the relatively smooth and filtered input data, it is known to be sensitive to outliers and data range extremes. In datasets with greater variability or different preprocessing methods, alternative scaling approaches such as standardization or robust scaling could be more appropriate. Future studies should assess the impact of different normalization strategies on clustering results, especially in settings with noisier or unfiltered sensor data."

- Lines 431-351: the authors perform 100 SOM runs per candidate cluster number and use the MCASD to select the optimal k. While this addresses compactness, there is no assessment of cluster stability. Please clarify whether variability across SOM runs was quantified (ARI or some cluster overlap metrics).

We thank the reviewer for this valuable comment. While we did not explicitly compute clustering overlap metrics such as the Adjusted Rand Index (ARI), our approach used the Multi-Cluster Average Standard Deviation (MCASD) inherently reflects variability across SOM runs. Specifically, MCASD quantifies the stability of cluster centers by averaging their standard deviation over multiple iterations. During preliminary testing, we observed that most datasets stabilized in terms of variability between 70 and 80 iterations. To ensure consistency and reproducibility, we adopted 100 runs per cluster number. This approach provided a reliable means to assess both compactness and relative stability of clusters in a computationally efficient manner. We have clarified this in the manuscript and added a note in the Limitations section (Section 3.5) to suggest the use of additional stability metrics like ARI in future work. The new text reads:

"While cluster variability was addressed using the Multi-Cluster Average Standard Deviation (MCASD) across 100 SOM runs, future studies may benefit from incorporating additional stability metrics such as the Adjusted Rand Index (ARI) or cluster overlap measures to better assess classification consistency."

- To enhance the clarity of the manuscript. The authors should consider including a workflow diagram summarizing the complete methodology.

We thank the reviewer for this helpful suggestion. To enhance clarity, we have added a workflow diagram (Figure 2) in Section 2.2 that summarizes the complete methodology, including the classification and validation steps. The diagram visually outlines the integration of EMI and NDVI data, the clustering process using SOMs and MCASD, and the post-hoc validation using yield and soil data.

The overall methodology of this study, including data, processing steps, and validation is summarized in Figure 2. This flowchart highlights the role of EMI and NDVI datasets in clustering process and the use of multi-year yield maps and soil samples for validation and refinement of the resulting management zones.

[Figure]

*Figure 2. Workflow diagram showing the integration of proximal (EMI) and remote sensing (NDVI) data for unsupervised clustering using MCASD and SOMs. Yield and soil datasets were used for post-hoc validation and refinement of management zones.*

- Lines 93-96: while NDVI is common vegetation index, it is well-known to saturate under high biomass or dense canopy conditions, which may limit its ability to capture within field variability during peak crop growth. The authors should justify why NDVI was selected over alternatives such as EVI or SAVI.

We thank the reviewer for the insightful comment. We acknowledge that NDVI can exhibit saturation under high biomass or dense canopy conditions, which may limit its sensitivity during peak growth. However, we used NDVI as: a) it can directly and reliably be derived from the PlanetScope sensor as well as from many other sensors (e.g. satellite-, aerial- and drone-based), b) the focus of our study was on capturing relative spatial variability within the field, not absolute vegetation productivity, and c) NDVI remains a widely accepted, validated, and simple index for evaluating vegetation vigour across phenological stages. In fact, other indices like EVI and SAVI can require specific calibration parameters (e.g., soil brightness correction factor or coefficients tied to aerosol resistance), which were not feasible to constrain accurately within our satellite dataset and field setting. We thus preferred to use NDVI, which does not require additional computation or calibration. We think that this makes for a simpler, ready to use, and transferrable approach. To avoid extending an already long manuscript, we would prefer to not provide additional justification in the manuscript.

- Lines 370-395: The initial presentation of yield maps provides useful spatial context. However, since the 2012 and 2013 data are acknowledged to be lower in quality, the authors should discuss whether these data were weighted differently or excluded from statistical validation to avoid introducing bias in zone validation.

We thank the reviewer for this important point. The 2012 and 2013 yield data were presented because they showed relevant spatial trends, despite lower data quality. To avoid introducing bias, these years were not weighted differently in the statistical validation. Instead, we relied on multi-year averages and year-by-year comparisons to assess the robustness of zone delineation. This clarification has now been added to the end of the yield data subsection (Lines 393–397). The new text reads:

"… they were retained for spatial context as they still exhibited consistent patterns with other years. These years were not weighted differently during validation analyses, and the potential influence of this lack of weighting was mitigated by evaluating multi-year trends and conducting year-by-year comparisons in the validation stage (see Section 3.4)."

- Given the spatial nature of EMI and NDVI data and the use of kriging interpolation, spatial autocorrelation is likely present in the dataset. While the current clustering is sound, the authors may consider briefly acknowledging the presence of spatial structure and its potential influence on post-hoc tests.

We thank the reviewer for this comment. We agree that kriging interpolation introduces spatial structure in the EMI and NDVI datasets, which can influence the assumptions underlying post-hoc statistical tests such as ANOVA and t-tests. While we did not explicitly correct for spatial autocorrelation, we believe its impact was mitigated through the use of multi-year yield data and non-interpolated soil sampling in the validation process. We have now included an explicit acknowledgment of this point in the Limitations section. The new text reads:

"This may influence statistical outcomes or lead to less spatially coherent clusters in some cases. Additionally, the use of kriging interpolation for EMI and NDVI datasets introduces spatial structure that may further affect the assumptions underlying post-hoc statistical tests."

---

## Author Response (AR1)

**Response to reviewers**

**Response to comments and suggestions from Reviewer 1**

**General comments**

The paper presents a relevant contribution to precision agriculture by coupling NDVI and EMI for management zones' delineation. The methodology is generally sound, especially the use of the SOM and MCASD for cluster optimisation. The study presents a robust workflow that could inform both research and practice. However, some aspects need clarification to improve generalisability and interpretability.

We thank the reviewer for the positive assessment, and are happy to provide the requested clarifications.

**Specific Comments**

- Lines 64-122: The review of EMI and NDVI is largely descriptive. It would be stronger if the authors synthesised how of the previous studies succeed or failed in integrating these data types. I suggest adding a short synthesis paragraph summarising what's missing in prior work and how this study fills the gap. **(DONE)**

We thank the reviewer for this helpful suggestion. In response, we have now included the following statement in section 1 (lines 118-125):

"... Overall, previous studies have made important contributions towards integrating EMI and NDVI data for improved management zone delineation (Corwin and Scudiero, 2019; Ciampalini et al., 2015). However, the results can be influenced by data resolution and acquisition timing as well as by local management and soil-plant interactions, with some studies suggesting that EMI alone can offer sufficient insights into soil patterns (Esteves et al., 2022; von Hebel et al., 2021). Nonetheless, the added value of NDVI holds unexplored potential due to the higher spatial and temporal resolution of recent satellite platforms (Breunig et al., 2020; Georgi et al., 2018).

- Lines 304- 309: the use of min-max scaling prior to clustering is appropriate for ensuring feature comparability. However, the authors should briefly justify this choice over alternatives (e.g., standardisation, robust scaling), especially given the potential presence of outliers in EMI and NDVI data. Min-max is sensitive to extreme values, which may distort the input space and affect cluster geometry in SOM. **(DONE)**

We thank the reviewer for pointing out this important consideration. In our study, EMI data were already filtered to remove outliers from a variety of sources. In particular, the combination of min-max filtering, histogram filtering, and ECa variation filtering effectively remove outliers from the EMI data, as shown in previous research. We therefore are confident that the resulting distribution of ECa values, combined with the use of z-transform normalization, is appropriate for a min-max scaling. Similarly, NDVI maps were pre-processed by PlanetScope to remove atmospheric artifacts, and we manually excluded data from periods with low vegetation signal (lines 452-457). Moreover, the extent of the area and the amount of pixels in the NDVI images assures that the distribution is free from outliers that would affect the min-max scaling. Nonetheless, we understand that this may not be the case in other areas or when different data sources are used. Thus, we now addressed these topics in section 3.5 (lines 693- 701), where the new text reads:

"Regarding data process, min-max scaling was a suitable method in this study due to the relatively smooth and filtered input data, both for EMI and NDVI. However, this scaling approach is known to be sensitive to outliers and data range extremes (Pedregosa et al., 2011). For datasets with greater variability or different preprocessing methods, alternative scaling approaches such as standardization or robust scaling could be more appropriate (de Amorim et al., 2023). Another factor was the proper application of data normalization prior to clustering, which was essential for obtaining meaningful results in this study (see Appendix A). Future studies should assess the impact of different scaling and normalization strategies on clustering outcomes, especially in settings with noisier or unfiltered sensor data."

- Lines 431-351: the authors perform 100 SOM runs per candidate cluster number and use the MCASD to select the optimal k. While this addresses compactness, there is no assessment of cluster stability. Please clarify whether variability across SOM runs was quantified (ARI or some cluster overlap metrics). **(DONE)**

We thank the reviewer for this valuable comment. While we did not explicitly compute clustering overlap metrics such as the Adjusted Rand Index (ARI), our approach used the Multi-Cluster Average Standard Deviation (MCASD) inherently reflects variability across SOM runs. Specifically, MCASD quantifies the stability of cluster centers by averaging their standard deviation over multiple iterations. During preliminary testing, we observed that most datasets stabilized in terms of variability between 70 and 80 iterations. To ensure consistency and reproducibility, we adopted 100 runs per cluster number. This approach provided a reliable means to assess both compactness and relative stability of clusters in a computationally efficient manner. We have clarified this in the manuscript and added a note in the Limitations section (Section 3.5, lines 704- 708) to suggest the use of additional stability metrics like ARI in future work. The new text reads:

"While cluster variability was addressed using the Multi-Cluster Average Standard Deviation (MCASD) across 100 SOM runs to a large extent, future studies may benefit from incorporating additional stability metrics such as the Adjusted Rand Index (ARI) or cluster overlap measures to better assess classification consistency."

- To enhance the clarity of the manuscript. The authors should consider including a workflow diagram summarizing the complete methodology. **(DONE)**

We thank the reviewer for this helpful suggestion. To enhance clarity, we have added a workflow diagram (Figure 2) in Section 2.2 (lines 191- 194) that summarizes the complete methodology, including the classification and validation steps. The diagram visually outlines the integration of EMI and NDVI data, the clustering process using SOMs and MCASD, and the post-hoc validation using yield and soil data.

The overall methodology of this study is summarized in Figure 2. This flowchart highlights the role of EMI and NDVI datasets in the clustering process and the use of multi-year yield maps and soil samples for validation and refinement of the resulting management zones. More details are provided in the subsequent sections.

[Figure]

*Figure 2. Workflow diagram showing the integration of proximal (EMI) and remote sensing (NDVI) data for unsupervised clustering using MCASD and SOMs. Yield and soil datasets were used for post-hoc validation and refinement of management zones.*

- Lines 93-96: while NDVI is common vegetation index, it is well-known to saturate under high biomass or dense canopy conditions, which may limit its ability to capture within field variability during peak crop growth. The authors should justify why NDVI was selected over alternatives such as EVI or SAVI. **(DONE)**

We thank the reviewer for the insightful comment. We acknowledge that NDVI can exhibit saturation under high biomass or dense canopy conditions, which may limit its sensitivity during peak growth. However, we used NDVI as: a) it can directly and reliably be derived from the PlanetScope sensor as well as from many other sensors (e.g. satellite-, aerial- and drone-based), b) the focus of our study was on capturing relative spatial variability within the field, not absolute vegetation productivity, and c) NDVI remains a widely accepted, validated, and simple index for evaluating vegetation vigour across phenological stages. In fact, other indices like EVI and SAVI can require specific calibration parameters (e.g., soil brightness correction factor or coefficients tied to aerosol resistance), which were not feasible to constrain accurately within our satellite dataset and field setting. We thus preferred to use NDVI, which does not require additional computation or calibration. We think that this makes for a simpler, ready to use, and transferrable approach. To avoid extending an already long manuscript, we would prefer to not provide additional justification in the manuscript.

- Lines 370-395: The initial presentation of yield maps provides useful spatial context. However, since the 2012 and 2013 data are acknowledged to be lower in quality, the authors should discuss whether these data were weighted differently or excluded from statistical validation to avoid introducing bias in zone validation. **(DONE)**

We thank the reviewer for this important point. The 2012 and 2013 yield data were presented because they showed relevant spatial trends, despite lower data quality. To avoid introducing bias, these years were not weighted differently in the statistical validation. Instead, we relied on multi-year averages and year-by-year comparisons to assess the robustness of zone delineation. This clarification has now been added to the end of the yield data subsection 3.1.3 (Lines 482–486). The new text reads:

"... they were retained for spatial context as they still exhibited consistent patterns with other years, and the maps successfully captured the general spatial yield trends and heterogeneity of the field. These years were not weighted differently during validation, and the potential influence of this lack of weighting was mitigated by evaluating multi-year trends and conducting year-by-year comparisons (see Section 3.4)."

- Given the spatial nature of EMI and NDVI data and the use of kriging interpolation, spatial autocorrelation is likely present in the dataset. While the current clustering is sound, the authors may consider briefly acknowledging the presence of spatial structure and its potential influence on post-hoc tests.

We thank the reviewer for this comment. We agree that kriging interpolation introduces spatial structure in the EMI and NDVI datasets, which can influence the assumptions underlying post-hoc statistical tests such as ANOVA and t-tests. While we did not explicitly correct for spatial autocorrelation, we believe its impact was mitigated through the use of multi-year yield data and non-interpolated soil sampling in the validation process. We have now included an explicit acknowledgment of this point in the Limitations section 3.5 (lines 718- 720). The new text reads:

"This may influence statistical outcomes or lead to less spatially coherent clusters in some cases. For instance, kriging interpolation introduces a spatial structure that may challenge the assumption of independence underlying post-hoc statistical tests."

**Response to comments and suggestions from Reviewer 2**

This paper proposes a proximal and remote sensing data harmonisation framework for input into a Self-organizing map (SOM)-based classification for determining field management zones. It is worthy of publication once the following points are considered and addressed.

We thank the reviewer for the positive evaluation. In the following, we describe how we have addressed the points raised by the reviewer.

1. Materials/Methods: The four sub-sections of section 2.2 need re-ordering to demonstrate the workflow: (1) EMI/EC data, (2) RS/NDVI data, (3) Yield data, (4) Soils data. As only the first two are inputs for the SOM/MCASD clustering. The second two are used to 'validate' and refine the clusters. **(DONE)**

We thank the reviewer for this helpful suggestion. We agree that reordering the subsection in Section 2.2 improves the clarity and logical flow of the methodology, particularly in distinguishing between input data for clustering (EMI, NDVI) and data used for validation and refinement (yield, soil). We have revised the manuscript accordingly by placing the subsections in the suggested order.

2. Materials/Methods: A table would be useful to summarise each of these four datasets and their use in the study. The table can list: (a) the period of collection (e.g., 2011-19 for yield data); (b) whether the patchCROP experiment was in operation or not, (c) data processing steps taken (e.g. kriging or some other interpolation, normalisation etc. – see also that stated in section 3), and (d) whether used for SOM/MCASD inputs or used for the (ANOVA-based) validation of SOM clusters (with subsequent merging of clusters) etc. **(DONE)**

We thank the reviewer for this very helpful suggestion. We agree that summarizing the role and processing of the four datasets enhances clarity. In response to a similar suggestion from Reviewer 1, we have added a workflow diagram (Figure 2 in the manuscript, Figure R1 below) that provides an overview of all data sources, their processing steps, and their roles in both the clustering and validation stages. We believe this figure addresses the intent of the suggested table in a more integrated and visual format, and improves the overall readability of the Materials and Methods section.

[Figure]

*Figure R1. Summary of workflow showing the integration of proximal (EMI) and remote sensing (NDVI) data for unsupervised clustering using MCASD and SOMs. Yield and soil datasets were used for post-hoc validation and refinement of management zones.*

3. Results: Maps and workflow narratives should be in this order: (1) EMI/EC data (Figs. 3, 4), (2) RS/NDVI data (Fig. 5), (3) Yield data (Fig. 2), (4) Soils data graphic (new), (5) SOM/MCASD clustering maps of EMI/RS plus refinements via yield/soils (Fig. 6). **(DONE)**

We thank the reviewer for this helpful recommendation regarding the logical flow of the Results section. We agree that reordering the narrative and associated figures to match the data processing workflow enhances clarity. As suggested, we have revised the Results section to follow this order: (1) EMI/ECa maps, (2) NDVI maps, (3) Yield data, and (4) SOM/MCASD clustering and refinement maps.

We decided not to add a new figure presenting the soil data, as we prefer to present them as part of the validation of the zonation.

4. Limitations: When describing the caveats to the methodology (section 3.5), refer to the new Table suggested in (2) for challenges due to different data collection timeframes, patchCROP, data processing, etc. **(DONE)**

We thank the reviewer for this observation. The methodological caveats related to differences in data collection timeframes, patchCROP implementation, and dataset-specific processing steps are already discussed in the Limitations section (Lines 666–696). While we did not include a table as initially suggested, we opted to incorporate a workflow diagram (Figure 2), which summarizes the sequence and role of each dataset. We believe that the combination of this figure and the existing discussion adequately addresses the reviewer's concern.

5. Limitations: What would be the likely consequences of using free, 10m resolution imagery from sentinel 2 say, to that used with the 3m resolution of Planetscope for the NDVI data? **(DONE)**

We thank the reviewer for raising this interesting point. The 3 m spatial resolution of PlanetScope imagery provided a more detailed representation of within-field variability, which was essential for our study's goal of delineating high-resolution management zones. In contrast, using 10 m resolution data from Sentinel-2 would likely result in a less detailed representation of the horizontal heterogeneity in NDVI, which could obscure narrow or patchy features, especially in highly heterogeneous fields like that of this study. This could reduce the sensitivity of the clustering algorithm to subtle spatial transitions and affect the precision of zone delineation. However, for larger fields or regions with less spatial variability, Sentinel-2 could be a valuable, freely available alternative. We have now included these considerations in section 3.5 (lines 670-680). The new text reads:

"Similarly, the NDVI dataset was limited to the 2019 growing season as a) PlanetScope imagery became accessible for this region only in 2019 and b) the subdivision of the field in differently cultivated patches from 2020 prevented the use of later satellite products. Nonetheless, the choice of PlanetScope imagery (3 m resolution) enabled to capture detailed within-field variability in NDVI, which was particularly important in this study area due to the spatial heterogeneity introduced by soil variation. If coarser-resolution imagery such as Sentinel-2 (10 m) were used instead, smaller-scale patterns in crop development or soil-related variation would have been less detectable due to spatial averaging. This could have reduced the effectiveness of the SOM clustering in identifying distinct management zones. However, for more homogeneous or large-scale fields, Sentinel-2 data could be a practical and freely accessible alternative (Kaya et al., 2025)."

6. Limitations: More on the sensitivity of the SOM-based clusters and their refinements using yield and soil information – from no data available to that available here (as shown in rows 3 and 4 in Fig.6). **(DONE)**

We thank the reviewer for raising this point. It is true that the availability of yield and soil data can influence the refinement of the SOM-based clusters. In our study, these datasets were used to validate and occasionally merge clusters that were not clearly different in terms of agronomic performance. While such validation improves the interpretability of the zones, we acknowledge that in cases where such data are not available or are sparse, the clustering process can still be applied—although some clusters may remain less interpretable. We have now included these considerations in section 3.5 (lines 708-714). The new text reads:

"The availability of yield and soil data supported the refinement of SOM-based clusters, enabling the merging of groups that were not agronomically distinct. These datasets helped to ensure that the final management zones were both data-driven and interpretable. However, in scenarios where such ground-truth data are limited or unavailable, the initial clusters may still offer useful insights, albeit with greater uncertainty in their agronomic interpretation. Thus, the presented post-hoc validation step added confidence in the results, but is not strictly required."

7. Limitations: For the clustering methods described (in the introduction) and the SOM method applied (p.6 to p.7) – none implicitly capture spatial effects, such as spatial autocorrelation. Further, the statistical analyses using ANOVAs/Tukey's HSD and t-tests are similarly non-spatial. What are the consequences of this? What methods could be applied for future work to investigate this? **(DONE)**

We thank the reviewer for this thoughtful observation. It is true that the clustering and statistical validation methods used in this study do not explicitly consider spatial autocorrelation, which may influence both the clustering output and the interpretation of statistical significance. While our use of multi-year yield data and soil samples helped support the robustness of the final zones, we recognize that spatial dependence remains an important factor. We have now included a paragraph in the Limitations section suggesting that future work could apply spatially-aware clustering methods or spatial statistical approaches to better account for this aspect (Section 3.5, lines 716-726). The new text reads:

"The SOM algorithm and the statistical methods used in this study (ANOVA, Tukey's HSD, and t-tests) do not explicitly account for spatial autocorrelation, which is inherently present in the interpolated geospatial datasets used here. This may influence statistical outcomes or lead to less spatially coherent clusters in some cases. For instance, kriging interpolation introduces a spatial structure that may challenge the assumption of independence underlying post-hoc statistical tests. However, the use of multi-year yield trends and high-resolution soil data helped reduce uncertainty in post-hoc validation. Future studies may benefit from incorporating spatially explicit methods, such as spatially constrained clustering, variogram-based diagnostics, or spatial ANOVA, to better account for spatial dependence during both classification and validation stages. In addition to these methodological considerations, future studies should focus on improving the temporal consistency of data collection and increasing the density and depth of soil sampling."

8. Limitations: Given all the above - something on the capture of uncertainty in the demarcation of the management zones for current and future work? **(DONE)**

We thank the reviewer for raising this relevant point. While this study did not explicitly quantify uncertainty in the delineation of management zones, we agree that this represents an important direction for future work. A sentence has been added to the Limitations section 3.5 (lines 726-728) to acknowledge this. The new text reads:

"... The quantification of uncertainty in management zone delineation could be also investigated, for example through ensemble clustering or by incorporating uncertainty from spatial inputs such as EMI interpolation."

9. Conclusion: More should be said on the choice made for the proximal sensing and the choice made for the satellite remote sensing. For the former, EMI/EC essentially does soil physics / structure / water, while for latter, NDVI does crop health. This is OK but what of the alternatives? For example, using indices from radar-based missions (e.g., sentinel 1) rather than imagery based missions (e.g., sentinel 2). Insights on how the choice of sensors will ultimately affect the SOM/MCASD clustering and resultant management zones would be useful. For example, in some cases, the precision management of soil water may be more of a focus than the precision management soil nutrients – each requiring specific sensing technologies, etc. Essentially expand discussions in the introduction (p.5-6) and conclusions. **(DONE)**

We thank the reviewer for this interesting point and we agree that the choice of sensing technology (e.g. optical, radar-based, or thermal imagery) can significantly influence the types of variability captured and the resultant management zones. It is also clear that the zones of management may depend on the type of management considered. However, we do not feel that a repetition of these points is fruitful in the conclusions. The objective of this study was to evaluate a harmonized, scalable workflow using widely available and well-established data sources: EMI for subsurface soil properties and NDVI for above-ground crop performance. Exploring the use of alternative sensors such as Sentinel-1 radar or hyperspectral imagery would indeed be valuable, but was beyond the scope of the present work. We prefer to not elaborated on this further in the conclusion section, as different sensors are now also addressed while discussing the limitations of the present study.

10. Consider changing the title to either: 'Combining Proximal and Satellite Remote Sensing Data for Improved Determination of Management Zones for Sustainable Crop Production' or 'Combining Electromagnetic Induction and Satellite Sensed NDVI Data for Improved Determination of Management Zones for Sustainable Crop Production' – the former is general, while the latter is specific. **(DONE)**

We thank the reviewer for the helpful suggestions regarding the title. We agree that a more descriptive title improves clarity and scope. Accordingly, we have revised the title to: "Combining Electromagnetic Induction and Satellite-based NDVI Data for Improved Determination of Management Zones for Sustainable Crop Production." This title reflects the specific sensing methods used in our study and aligns with the reviewer recommendation.